# Structure-based analysis of CysZ-mediated cellular uptake of sulfate

**Zahra Assur Sanghai**[1,2], **Qun Liu**[3], **Oliver B Clarke**[2], **Meagan Belcher-Dufrisne**[1], **Pattama Wiriyasermkul**[4], **M Hunter Giese**[1,2], **Edgar Leal-Pinto**[5,6], **Brian Kloss**[7], **Shantelle Tabuso**[7], **James Love**[7], **Marco Punta**[8], **Surajit Banerjee**[9], **Kanagalaghatta R Rajashankar**[9], **Burkhard Rost**[10], **Diomedes Logothetis**[5,6], **Matthias Quick**[4,11], **Wayne A Hendrickson**[1,2,7], **Filippo Mancia**[1]*

[1]Department of Physiology and Cellular Biophysics, Columbia University, New York, United States; [2]Department of Biochemistry and Molecular Biophysics, Columbia University, New York, United States; [3]Biology Department, Brookhaven National Laboratory, Upton, United States; [4]Center for Molecular Recognition, Department of Psychiatry, Columbia University, New York, United States; [5]Department of Physiology and Biophysics, Virginia Commonwealth University School of Medicine, Richmond, United States; [6]Department of Pharmaceutical Sciences, School of Pharmacy, Bouvé College of Health Sciences, Northeastern University, Boston, United States; [7]New York Structural Biology Center, New York, United States; [8]Centre for Evolution and Cancer, The Institute of Cancer Research, London, United Kingdom; [9]Department of Chemistry and Chemical Biology, Cornell University, NE-CAT, Argonne, United States; [10]Department of Informatics, Technical University of Munich, Munich, Germany; [11]Division of Molecular Therapeutics, New York State Psychiatric Institute, New York, United States

*For correspondence:
fm123@columbia.edu

**Competing interests:** The authors declare that no competing interests exist.

**Abstract** Sulfur, most abundantly found in the environment as sulfate ($SO_4^{2-}$), is an essential element in metabolites required by all living cells, including amino acids, co-factors and vitamins. However, current understanding of the cellular delivery of $SO_4^{2-}$ at the molecular level is limited. CysZ has been described as a $SO_4^{2-}$ permease, but its sequence family is without known structural precedent. Based on crystallographic structure information, $SO_4^{2-}$ binding and flux experiments, we provide insight into the molecular mechanism of CysZ-mediated translocation of $SO_4^{2-}$ across membranes. CysZ structures from three different bacterial species display a hitherto unknown fold and have subunits organized with inverted transmembrane topology. CysZ from *Pseudomonas denitrificans* assembles as a trimer of antiparallel dimers and the CysZ structures from two other species recapitulate dimers from this assembly. Mutational studies highlight the functional relevance of conserved CysZ residues.
DOI: https://doi.org/10.7554/eLife.27829.001

## Introduction

Sulfur has a central role in many cellular processes across all kingdoms of life. It is a vital component of several essential compounds, including the sulfur-containing amino acids cysteine and methionine, in prosthetic groups such as the Fe-S clusters, as well as vitamins and micronutrients such as biotin (vitamin H), thiamine (vitamin B1) and lipoic acid, and in coenzymes A and M (*Barton, 2005*). Whilst mammals obtain the majority of the necessary sulfur-containing metabolites directly from the diet, plants, fungi, and bacteria are able to assimilate and utilize sulfur from organic and inorganic sources (*Barton, 2005*). Sulfate ($SO_4^{2-}$) is the most abundant source of sulfur in the environment and its

utilization is contingent upon its entry into the cell (*Kertesz, 2000*). In certain fungi, and prokaryotes, once internalized, $SO_4^{2-}$ is first reduced to sulfite ($SO_3^{2-}$), and then further to sulfide ($S^{2-}$), a form that can be used by the cell (*Kredich et al., 1979*) (*Figure 1—figure supplement 1*). In *Escherichia coli* and other gram-negative bacteria, the culmination of the aforementioned sulfate assimilatory (also known as reductive) pathway is the formation of cysteine by the addition of $S^{2-}$ to O-acetylserine by cysteine synthase, followed by the synthesis of methionine from homocysteine (*Kredich, 1971*) (*Figure 1—figure supplement 1*).

In prokaryotes, the entry of $SO_4^{2-}$ into the cell is mediated by four known families of dedicated transport systems: the ABC sulfate transporter complexes SulT or CysTWA, the SulP family of putative SLC13 sodium:sulfate or proton:sulfate symporters or SLC26 solute:sulfate exchangers, the phosphate transporter-like CysP/PitA family, and the CysZ family classified as $SO_4^{2-}$ permeases (*Aguilar-Barajas et al., 2011*; *Hryniewicz et al., 1990*; *Kertesz, 2001*; *Loughlin et al., 2002*; *Mansilla and de Mendoza, 2000*; *Sirko et al., 1995*). CysZ family members are 28–30 kDa bacterial inner-membrane proteins found exclusively in prokaryotes with no apparent homology to any of the established channel or transporter folds, and are scarcely studied in the literature (*Zhang et al., 2014*). The *cysZ* gene owes its name to its presence in the cysteine biosynthesis regulon. In two reports from thirty years ago, an *E. coli* K-12 strain with a *cysZ* deletion showed a severe impairment in its ability to accumulate $SO_4^{2-}$ and was not viable in sulfate-free media without an alternate sulfur source such as thiosulfate ($S_2O_3^{2-}$) (*Britton et al., 1983*; *Parra et al., 1983*). More recently, a third report studying the functional properties of CysZ, concluded that the protein from *E. coli* functions as a high affinity, highly specific pH-dependent $SO_4^{2-}$ transporter, directly regulated by the toxic, assimilatory pathway intermediate, $SO_3^{2-}$ (*Zhang et al., 2014*).

To investigate the role of CysZ in cellular sulfate uptake at a molecular level, we have undertaken an approach that combines structural and functional studies. To this end, we determined the crystal structures of CysZ from three species, *Idiomarina loihiensis (Il; IlCysZ)*, *Pseudomonas fragi (Pf; PfCysZ)*, and *Pseudomonas denitrificans (Pd; PdCysZ)*, and characterized CysZ function in purified form, predominantly in reconstituted proteoliposomes, but also in cells, and to a lesser extent, in planar lipid bilayers. Combining the structural information from the three orthologs reveals that CysZ features a novel protein fold that assembles as oligomers with an inverted transmembrane topology. This arrangement can be understood as being derived from trimers of dimers akin to the hexameric assembly captured in one of the structures. Interpreting the functional data in a structural context has allowed us to formulate a mechanistic model for CysZ-mediated $SO_4^{2-}$ translocation across the bacterial cytoplasm membrane.

Both the structures and the functional properties of CysZ proteins are distinct from those of any known membrane transporter or ion channel. Besides not resembling other transporter structures, CysZ mediates sulfate flux into cells or proteoliposomes without coupling to ion gradients, partner proteins, or exogenous energy sources such as ATP. These distinctive properties make CysZ appealing as a model system for studies of biophysical principles of membrane protein biogenesis and transmembrane ion passage.

## Results

### Structure determination of CysZ

Following a structural genomics approach aimed at crystallization for structural analysis, we cloned and screened a total of 63 different bacterial homologs of CysZ for high-level expression and stability in detergents (*Love et al., 2010*; *Mancia and Love, 2011*). Crystal structures were determined for CysZ from three organisms. Chronologically, the structure of *IlCysZ* was the first solved, to 2.3 Å resolution in space group *C2* by SAD, initially based on a single selenate ion bound to the protein and subsequently also by selenomethione derivatization (SeMet) SAD, and multi-crystal native SAD (*Liu et al., 2012*). The structure of *PfCysZ* was solved second, to 3.5 Å resolution, with crystals also belonging to space group *C2*. Although *PfCysZ* and *IlCysZ* share 42% sequence identity, molecular replacement failed to find a convincing solution, and we instead used SeMet-derivatized *PfCysZ* to obtain phase information by multi-crystal SeMet SAD. Third, we determined the structure of *PdCysZ*, which crystallized in multiple forms belonging to space groups *P6₃*, *P4₁22* and *P2₁2₁2₁*, revealing the same architecture and oligomeric assembly each time (*Figure 1—figure supplement 2b*).

*Pd*CysZ structures in the *P6₃* and *P2₁2₁2₁* lattices each contain an entire hexamer in their asymmetric units, whereas a molecular diad coincides with a crystallographic axis in the *P4₁22* lattice. We focused our analysis on the best of these (3.4 Å resolution in *P6₃*). The location of the selenium sites was obtained from a SeMet SAD data set, and the structure was solved by combining the resulting SAD phases with those from a molecular replacement solution obtained by positioning the *Pf*CysZ model (75% sequence identity) onto SeMet fiducials in the initial electron density map (*Table 1*, *Figure 1—figure supplement 2a*).

## The hexameric structure of *Pd*CysZ

The refined structure of *Pd*CysZ comprises an entire hexamer of near-perfect D3 symmetry. Antiparallel pairs of protomers arrange together as a trimer of dimers (*Figure 1a*), with the three-fold axis oriented perpendicular to the plane of the putative membrane and three two-fold axes between dimers of the hexamer. Both the periplasmic and cytoplasmic faces of the hexamer are essentially identical by symmetry, resulting in a dual-topology assembly for *Pd*CysZ. The hexamer has a

**Table 1.** Crystallographic data and refinement statistics.

| Data collection | Native *Il*CysZ | *Il*CysZ w/SeO₄²⁻ (SAD) | SeMet *Pf*CysZ (six crystals, refinement) | SeMet *Pf*CysZ (15 crystals, SAD) | Native *Pd*CysZ (MR-SAD) |
|---|---|---|---|---|---|
| Beamline | NSLS X4A | NSLS X4A/C | APS 24-ID-E | APS 24-ID-C | APS 24-ID-C |
| Space group | C2 | C2 | C2 | C2 | P6₃ |
| Cell dimensions: | | | | | |
| a, b, c (Å) | 128.9, 82.0, 100.4 | 128.9, 81.9, 100.3 | 172.29, 56.9, 96.17 | 172.35, 56.9, 96.31 | 225.13, 225.13, 96.62 |
| α, β, γ (°) | 90, 125.1, 90 | 90, 125.1, 90 | 90, 91.43, 90 | 90, 91.33, 90 | 90, 90, 120 |
| $Z_a$ | 2 | 2 | 2 | 2 | 6 |
| Wavelength | 1.7432 | 0.96789 | 0.97890 | 0.97890 | 1.0230 |
| Bragg spacings (Å) | 30–2.30 | 50–2.10 | 40–3.20 | 40–3.50 | 86–3.40 |
| $R_{merge}$ | 0.047 (0.257) | 0.046 (0.417) | 0.136 (6.506) | 0.135 (2.13) | 0.095 (2.89) |
| I / $\sigma_I$ | 29.4 (7.9) | 25.0 (1.9) | 12.3 (0.7) | 24.2 (2.6) | 22.4 (1.2) |
| Completeness (%) | 99.9 (99.9) | 99.9 (100.0) | 99.8 (99.3) | 99.9 (100.0) | 100.0 (100.0) |
| Multiplicity | 7.3 (7.2) | 7.6 (7.6) | 27.2 | 72.8 | 19.7 |
| Refinement: | | | | | |
| Resolution (Å) | 2.30 | | 3.50 | | 3.40 |
| No. of reflections | 38075 | | 20741 | | 37251 |
| $R_{work}/R_{free}$ | 0.200/0.238 | | 0.297/0.339 | | 0.245/0.289 |
| No. of atoms: | | | | | |
| Protein | 3679 | | 3428 | | 11415 |
| Ligand/ion | 249 | | 0 | | 400 |
| Water | 162 | | 0 | | 0 |
| Average B-factors (Å²) Protein Ligand/Ion Water | 49.4 48.4 77.3 45.3 | | 198.9 198.9 - - | | 178.58 179.06 164.88 - |
| Bond Ideality (r.m.s.d.): | | | | | |
| Bond lengths (Å) | 0.006 | | 0.006 | | 0.010 |
| Bond angles (°) | 0.894 | | 1.133 | | 0.96 |
| Ramachandran Analysis: Favored (%) | 99.0 | | 99.2 | | 96.27 |
| PDB accession code | 3TX3 | | 6D79 | | 6D9Z |

Values in parentheses are from the highest resolution shell. $R_{free}$ was calculated using 5% of data excluded from refinement.
DOI: https://doi.org/10.7554/eLife.27829.002

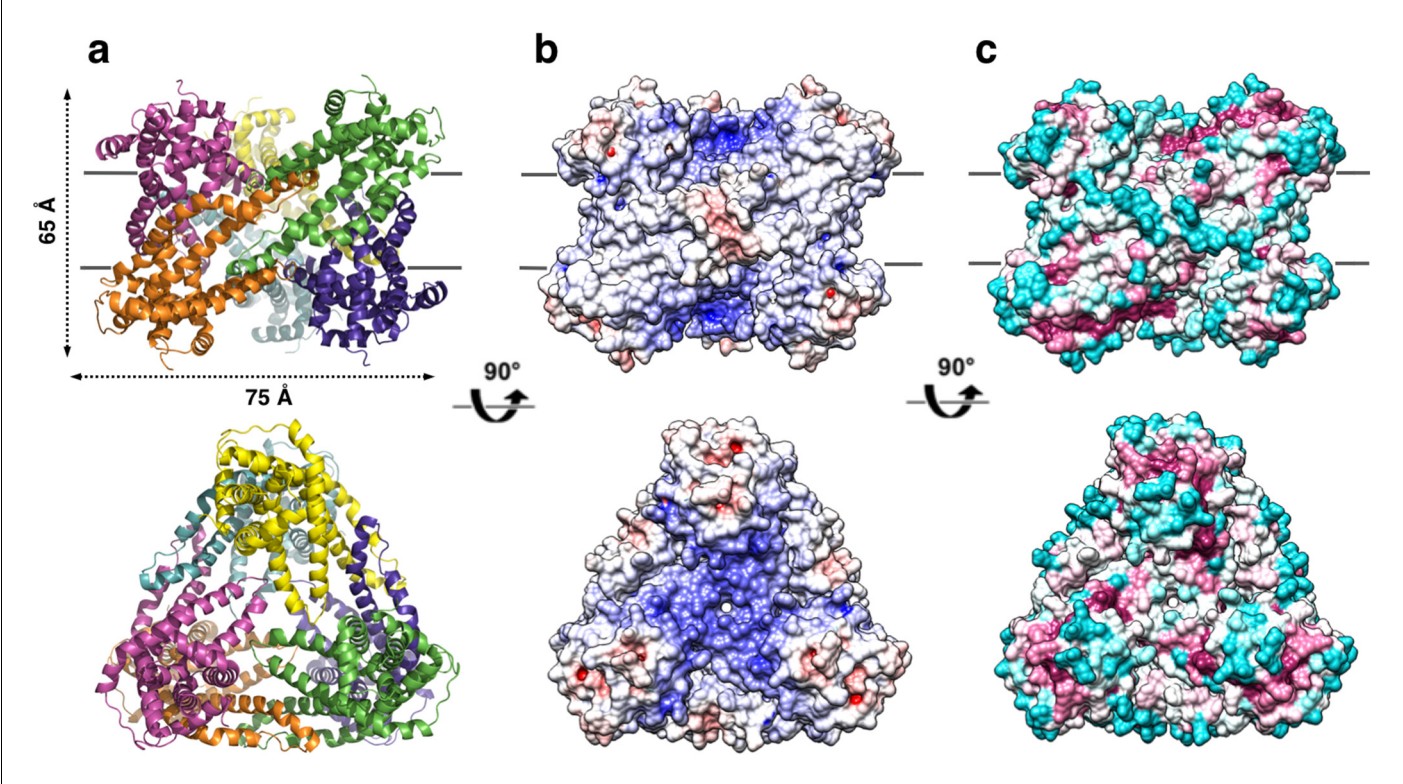

**Figure 1.** Overall structure of the *P. denitrificans* CysZ (*Pd*CysZ) hexamer. (a) Side and top views of the hexamer as a ribbon diagram with each protomer chain colored differently. The approximate dimensions of the hexamer marked in Å. (b) Side and top views represented by surface electrostatics as calculated by APBS, with negative and positive surface potential represented in red and blue respectively. (c) Side and top views representing conservation of residues as calculated by ConSurf, with maroon being most conserved to cyan being least conserved.
DOI: https://doi.org/10.7554/eLife.27829.003

The following figure supplements are available for figure 1:

**Figure supplement 1.** Schematic of assimilatory sulfate reduction in bacteria for cysteine biosynthesis.
DOI: https://doi.org/10.7554/eLife.27829.004

**Figure supplement 2.** Representative electron density of the crystal structures of *Il*CysZ, *Pf*CysZ and *Pd*CysZ, and the different crystal forms observed for *Pd*CysZ.
DOI: https://doi.org/10.7554/eLife.27829.005

triangular face of equal sides measuring approximately 75 Å, with the perpendicular span of about 65 Å. The interaction of the six protomers results in a total buried surface area (*Krissinel and Henrick, 2007*) of 5,700 Å$^2$. A surface electrostatic representation reveals a hydrophobic belt along the mid-section of the hexamer when viewed from its side, outlining the orientation of CysZ in the lipid bilayer (*Czodrowski et al., 2006*; *Dolinsky et al., 2007*; *Dolinsky et al., 2004*) (*Figure 1b*). A surface representation of the sequence conservation, calculated by analysis of multiple sequence alignments (MSA) (*Ashkenazy et al., 2016*; *Glaser et al., 2003*) highlights the regions of invariance in the sequence and in turn, the areas on the molecule that are most likely to have structural and functional importance (*Figure 1c*).

The CysZ protomer is an alpha-helical integral membrane protein with two long transmembrane (TM) helices (H2b and H3a) and two pairs of shorter helices (H4b-H5a and H7-H8) that insert only partially into the membrane (hemi-penetrating), forming a funnel or tripod-like shape within the membrane (*Figure 2a,b*). The protein has an extra-membranous hydrophilic 'head', comprising an iris-like arrangement of the two short helices, H1 and H6, and kinked helices H3b, H4a, and H5b. The amino and carboxyl termini are also located in this region (*Figure 2a,b*). Helices H4b and H5a lie partly inserted in the membrane, with the turn between the two pointing in towards the three-fold axis at the center of the hexamer. CysZ amino-acid sequences from different organisms are very similar; for example, those of *Pd*CysZ and *E. coli* CysZ (*Ec*CysZ) are 40.5% identical and those of

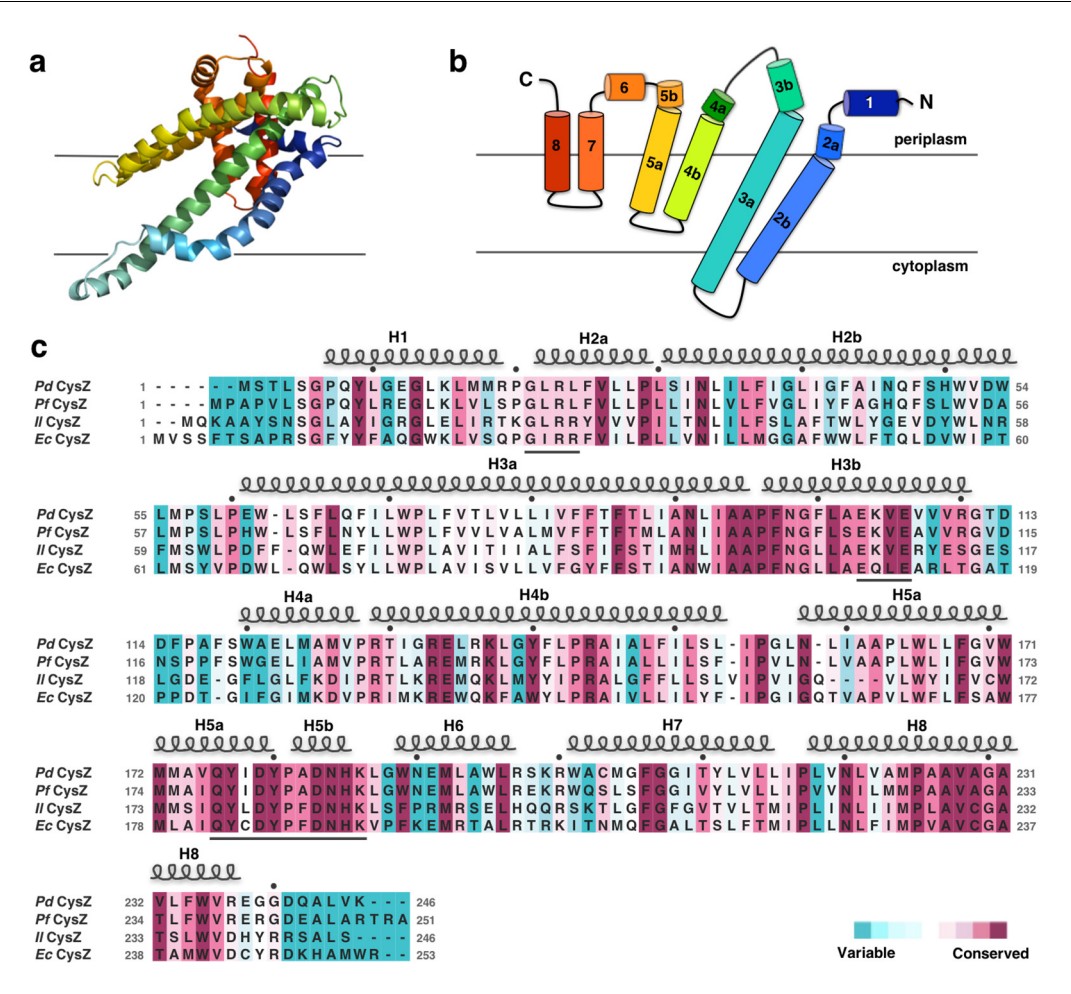

**Figure 2.** Structure and topology diagram of the *Pd*CysZ protomer, with sequence alignment and conservation. (a) Ribbon representation of the *Pd*CysZ protomer colored in rainbow colors from N (blue) to C terminus (red), viewed from within the plane of the membrane, shown in the same orientation as the protomer drawn in green in *Figure 1a*. (b) Topology diagram of the *Pd*CysZ protomer with helices marked from 1 to 8. Helices H2b and H3a are transmembrane helices, whereas helices H4-H5 and H7-H8 are hemi-penetrating helical hairpins, only partially inserted into the membrane. (c) Sequence alignment of *E. coli* CysZ (*Ec*CysZ), *P. denitrificans* CysZ (*Pd*CysZ), *P. fragi* CysZ (*Pf*CysZ), and *I. loihiensis* CysZ (*Il*CysZ). Residues are colored based on conservation, with maroon being most conserved and cyan least conserved, as calculated by ConSurf using a sequence alignment of 150 non-redundant sequences from the CysZ family as input. Spirals above residues mark the extent of the helical segments based on the atomic structure of *Pd*CysZ with helices numbered H1-H8; letters mark residue identities; black dots above identify every tenth residue (modulo 10) in the *Pd*CysZ sequence and black underlines mark functionally relevant motifs discussed in the text.

DOI: https://doi.org/10.7554/eLife.27829.006

*Ec*CysZ and our three structures have 30.0% of their residues exactly in common (*Figure 2c*). Helix boundaries are also essentially the same in the structures of *Pd*CysZ, *Il*CysZ and *Pf*CysZ (*Figure 2c*).

## The dimeric assembly of *Il*CysZ and *Pf*CysZ

Unlike *Pd*CysZ, both *Il*CysZ and *Pf*CysZ crystallize as dimers (*Figure 3a,b*), in agreement with their different behavior in detergent-containing solution. Indeed, size-exclusion chromatography runs of the three species of CysZ in the same buffer and detergent conditions, show mono-disperse peaks eluting at 13.42 ml (*Il*CysZ), 13.95 ml (*Pf*CysZ) and 12.56 ml (*Pd*CysZ) (*Figure 3—figure supplement*

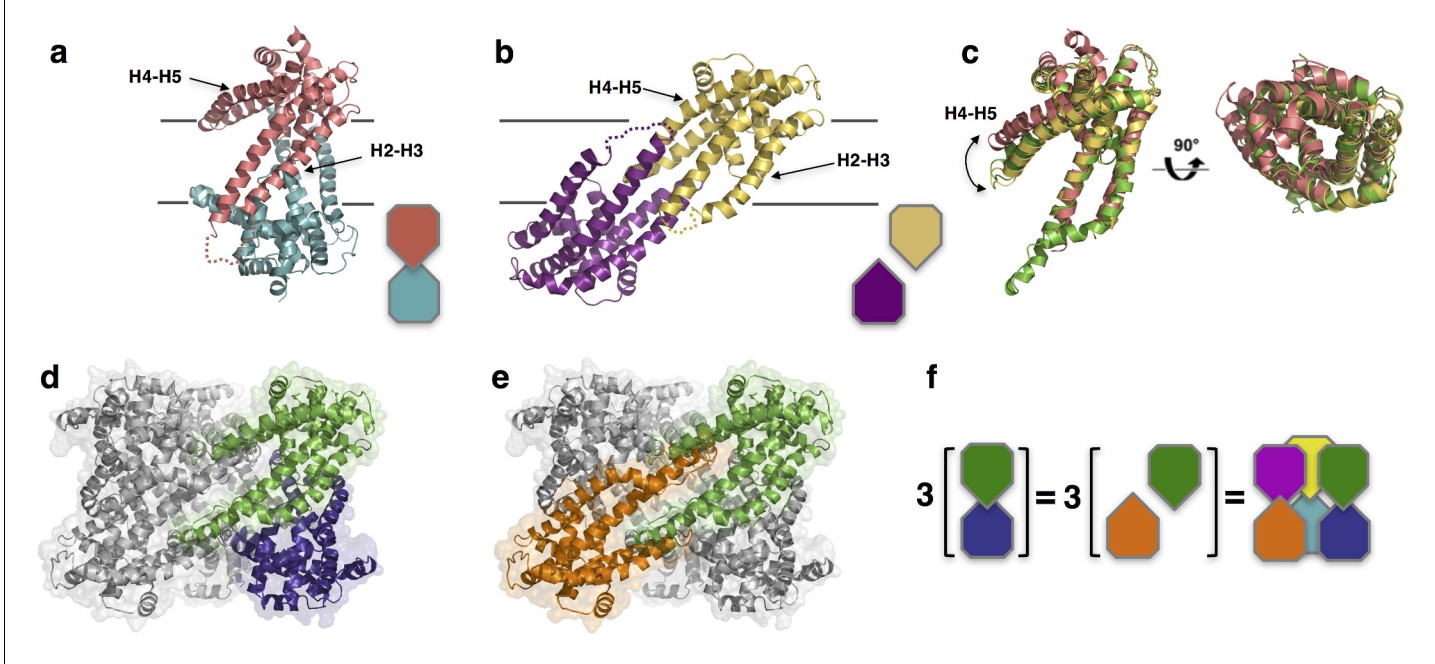

**Figure 3.** Structures of *Il*CysZ and *Pf*CysZ, with comparison to *Pd*CysZ. (a) Ribbon diagram of the structure of CysZ from *I. loihiensis* (*Il*CysZ) at 2.3 Å. Protomers of the dimer are colored in salmon pink and teal blue, arranged in a head-to-tail conformation in the membrane, with helical hairpins H2-H3 and H4-H5 labeled for clarity. The dimer interface of *Il*CysZ involves helices H2-H3. (b) Ribbon diagram of the structure of CysZ from *P. fragi (Pf*CysZ) at 3.2 Å. The protomers of the dual topology dimer in the membrane are colored gold and purple, with helical hairpins H2-H3 and H4-H5 again labeled. The dimer interface here involves the interaction of helices H4-H5 of each protomer. (c) Side and top views of the superposition of the three different protomers from *Pd*CysZ (green), *Il*CysZ (pink) and *Pf*CysZ (yellow) after aligning helices H1-H3, with a curved arrow indicating the flexible nature of helices H4-H5. d, e. The same dimer interfaces observed in *Il*CysZ (a) and *Pf*CysZ (b) observed in the hexameric assembly of *Pd*CysZ, as highlighted in green and blue (d) and green and orange (e). (f) Schematic representation showing how three copies of the dimeric protomers of *Il*CysZ (green and blue, left) and of *Pf*CysZ (orange and green, center), can coexist in and each recapitulate the hexameric assembly of *Pd*CysZ.
DOI: https://doi.org/10.7554/eLife.27829.007

The following figure supplements are available for figure 3:

**Figure supplement 1.** Size-exclusion chromatography of CysZ shows a mono-disperse elution profile for each of the three species purified - *Pd*CysZ, *Pf*CyZ and *Il*CysZ.
DOI: https://doi.org/10.7554/eLife.27829.008

**Figure supplement 2.** Surface electrostatics, hydrophobicity and site-directed fluorescence labeling of cysteine mutants located on H4 of *Il*CysZ.
DOI: https://doi.org/10.7554/eLife.27829.009

*1*). This result is consistent with a different and smaller oligomeric state of *Il*CysZ and *Pf*CysZ (similar retention volume) compared to *Pd*CysZ (eluting 1 ml ahead).

The protomers of the *Il*CysZ dimer are arranged in a head-to-tail antiparallel association, with helices H4b-H5a protruding at an angle that is nearly parallel to the putative plane of the membrane (*Figure 3a and* -figure supplement 2a,b). The *Pf*CysZ dimer is also arranged in an antiparallel orientation but with a different dimer interface (*Figure 3b*). In the *Pf*CysZ structure, helices H4b-H5a together form a narrower angle with helices H2 and H3, and tuck-in closer to the rest of the molecule (*Figure 3b*). The resulting dumbbell-shaped *Pf*CysZ dimer is predicted, by OPM/PPM (Orientation of Proteins in Membranes) (*Lomize et al., 2012*), to lie in the membrane at a 31° tilt to the perpendicular, in agreement with the position of its central hydrophobic belt, as revealed by surface-electrostatics calculations (*Figure 3—figure supplement 2e,f*). The dimer interfaces of *Il*CysZ and *Pf*CysZ bury 780 Å$^2$ and 1136 Å$^2$ of surface area respectively (*Krissinel and Henrick, 2007*)

The individual protomers of CysZ from all three species adopt the same topology and fold, and superpose well with an overall pairwise root mean squared deviation (r.m.s.d) of ~2.5 Å (*Figure 3c*). The greatest variation between the protomers of each structure is seen in the orientation of

helices H4b-H5a with respect to the TM helices H2 and H3 (*Figure 3c*), which seem to be the most conformationally flexible with respect to the rest of the molecule.

Comparison of the structures of *Il*CysZ and *Pf*CysZ with that of *Pd*CysZ revealed that these two distinctive dimeric structures are both represented in the hexameric one. Indeed, the *Il*CysZ dimer (*Figure 3a*) resembles the vertically arranged pair of protomers in the *Pd*CysZ hexamer (*Figure 3d*) at each of the vertices of the triangular structure. On the other hand, the *Pf*CysZ dimer (*Figure 3b*) resembles the transverse pair of protomers lying at a tilt, just as was predicted by OPM (*Lomize et al., 2012*), along the side of the *Pd*CysZ hexamer when viewed from inside the plane of the membrane (*Figure 3e*). In essence, the *Pd*CysZ hexamer can be seen as a trimer of either *Il*CysZ or *Pf*CysZ dimers (*Figure 3f*).

## Inverted transmembrane assembly of CysZ

To validate the inverted transmembrane assembly of CysZ observed in all our structures, we performed disulfide crosslinking assays on engineered cysteine mutants of CysZ designed to capture the antiparallel dimer, utilizing isolated membranes. Disulfide-trapping experiments of the transverse dimer, performed by crosslinking a pair of mutants (L161C-A164C) on helix H5 of *Pf*CysZ, as well as the corresponding pair in *Il*CysZ (V157C-Q163C) confirm the dimer interface observed in the structures of *Pf*CysZ and *Il*CysZ, and, as a consequence, the antiparallel assembly of the two proteins (*Figure 3—figure supplement 2f,g,h*).

To further validate the orientation of CysZ in the membrane, we performed a cysteine accessibility scan experiment, mapping residues expected to be located outside the lipid bilayer, by fluorescence labeling of the thiol groups of various single cysteine mutants with a membrane-impermeable dye. Membrane fractions isolated from recombinant cultures expressing *Il*CysZ cysteine mutants introduced at positions along the edge of helix H4 predicted to be solvent accessible based on our structure, were indeed labeled with a membrane-impermeable fluorescent thiol-specific maleimide dye (*Figure 3—figure supplement 2c,d*).

## Functional characterization of CysZ

To characterize the functional properties of CysZ, we performed (i) radiolabeled $[^{35}S]O_4^{2-}$ flux measurements in intact *E. coli* cells and in proteoliposomes containing reconstituted CysZ and (ii) radiolabeled ($[^{35}S]O_4^{2-}$) binding assays of purified CysZ in detergent solution. We also used single-channel electrophysiological recordings in planar lipid bilayer reconstituted with CysZ.

First, we compared the time course of $SO_4^{2-}$ accumulation in an *E. coli cysZ* knockout strain (*E. coli* K-12 JW2406-1, CysZ⁻) (*Baba et al., 2006*) with that in the wild-type (WT, *E. coli* K12 BW25113, *cysZ⁺*) strain. CysZ⁻ cells, after growth in minimal media and 12 hr of sulfate starvation, showed significantly diminished $SO_4^{2-}$ uptake when compared to the WT strain (*Figure 4a*), consistent with previous results (*Parra et al., 1983*). WT cells showed fast accumulation of 320 μM $[^{35}S]O_4^{2-}$ that was linear over a period of 3 min, whereas uptake of $[^{35}S]O_4^{2-}$ by the CysZ⁻ strain started to plateau after about 1 min. This low level of sulfate accumulation by the CysZ⁻ strain could be attributed to the other endogenous sulfate transport systems present in the bacteria (for example ABC transporter and SulP).

To assess direct CysZ-mediated $SO_4^{2-}$ flux without the potential interference of other native sulfate-transporting systems observed in the bacterial expression host, we henceforth conducted $[^{35}S]O_4^{2-}$ uptake experiments with purified CysZ reconstituted in proteoliposomes. Consistent with the cell uptake studies, the three orthologs (*Pd*CysZ, *Pf*CysZ and *Il*CysZ) all mediated the accumulation of 500 μM $SO_4^{2-}$ into proteoliposomes in a manner virtually indistinguishable from one another. Uptake was time-dependent and occurred rapidly with a peak at about 15 s (*Figure 4b*).

Testing the previously described inhibitory effect of $SO_3^{2-}$ (*Zhang et al., 2014*), we measured uptake of 500 μM $[^{35}S]O_4^{2-}$ in the presence of 500 μM $Na_2SO_3$ (*Figure 4—figure supplement 1a*). All three species of CysZ displayed $SO_3^{2-}$ sensitivity and their activity was virtually indistinguishable from the $SO_4^{2-}$ accumulation observed in control liposomes lacking CysZ (*Figure 4—figure supplement 1a*). Measuring uptake of 1 mM $[^{35}S]O_4^{2-}$ in the presence of increasing concentrations of $SO_3^{2-}$ showed that ~3 mM $SO_3^{2-}$ resulted in half-maximum inhibition ($IC_{50}$) of $SO_4^{2-}$ uptake under those experimental conditions (*Figure 4—figure supplement 1b*). The further kinetic characterization of the three CysZ isoforms in proteoliposomes (*Figure 4c*) revealed that the affinity for $SO_4^{2-}$ transport

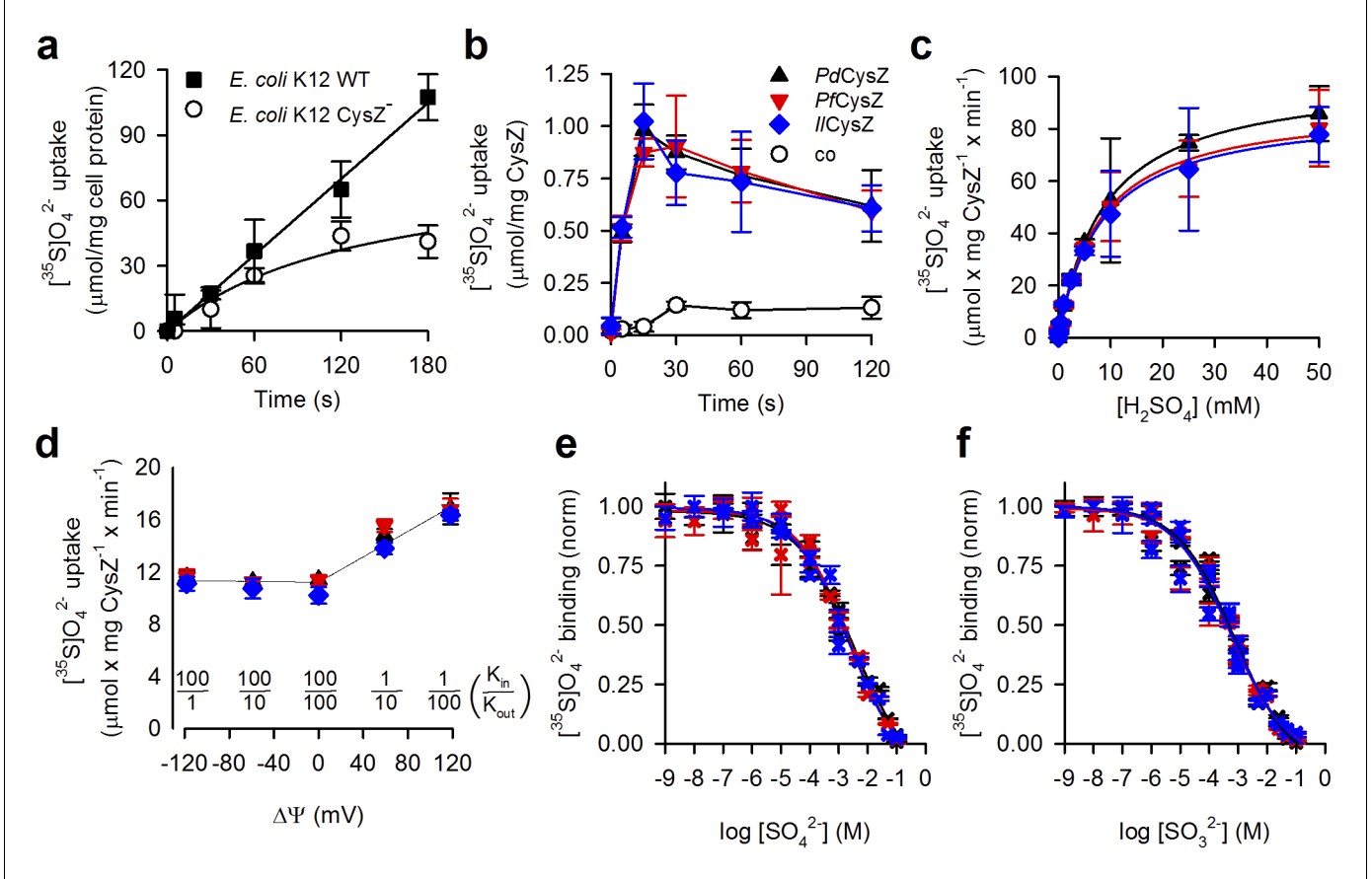

**Figure 4.** Functional characterization of CysZ. (a) Time course of [$^{35}$S]O$_4^{2-}$ uptake (320 μM) by CysZ$^+$ (strain BW25113) or CysZ$^-$ (strain JW2406-1)*E. coli* K-12 cells (n = 3). (b) Time course of [$^{35}$S]O$_4^{2-}$ uptake (500 μM) by CysZ-containing proteoliposomes (▲, *Pd*CysZ; ▼, *Pf*CysZ; ◆, *Il*CysZ) or by control liposomes (š). (c) Kinetics of [$^{35}$S]O$_4^{2-}$ uptake by CysZ. Initial rates of [$^{35}$S]O$_4^{2-}$ uptake were measured for 5 s with [$^{35}$S]O$_4^{2-}$ concentrations ranging from 0.5 to 50 mM. (d) Effect of the membrane potential (ΔΨ) on the initial rate of 1 mM [$^{35}$S]O$_4^{2-}$ uptake measured for a period of 5 s. ΔΨ of different polarity was imposed with the K$^+$-ionophore valinomycin in CysZ-containing proteoliposomes that were pre-loaded with different concentrations of KCl and assayed in external buffer of different KCl concentrations (equiosmolar cis/trans conditions) as indicated. The valinomycin-generated K$^+$ flux produced a KCl in/out concentration difference-dependent ΔΨ that was calculated using the Nernst equation ($N_e = \frac{RT}{zF} \ln \frac{[K^+]_{out}}{[K^+]_{in}}$). Data in panel (b–d) depict the means ± S.E.M. of 3 independent measurements performed in triplicate. (e, f) Equilibrium binding of [$^{35}$S]O$_4^{2-}$ measured with the SPA. (e) Isotopic dilution of 100 μM [$^{35}$S]O$_4^{2-}$ with non-labeled SO$_4^{2-}$ (sodium salt) or (f) competition of 100 μM [$^{35}$S]O$_4^{2-}$ with non-labeled SO$_3^{2-}$ (sodium salt) reveals the EC$_{50}$ or IC$_{50}$ (concentration yielding half-maximum displacement or inhibition, respectively). See text for kinetic constants. Data in (e and f) are shown as mean ± S.E.M. of ≥6 measurements and subjected to global fitting in Prism seven and kinetic constants reflect the mean ± S.E.M. of the fit. The use of color for symbols and bars in panel (b–f) is consistent.

DOI: https://doi.org/10.7554/eLife.27829.010

The following figure supplement is available for figure 4:

**Figure supplement 1.** Sulfite inhibition, effects of dissipated ion gradients, and ion conductance activities of CysZ.

DOI: https://doi.org/10.7554/eLife.27829.011

($K_m$) by CysZ is about 8 mM (*Pd*CysZ: 8.69 ± 0.35 mM; *Pf*CysZ: 7.86 ± 0.72 mM; *Il*CysZ: 8.03 ± 0.68 mM), while the maximum velocity of transport ($V_{max}$) is about 90 μmol SO$_4^{2-}$ x mg CysZ$^{-1}$ x min$^{-1}$ (*Pd*CysZ: 100.4 ± 1.4; *Pf*CysZ: 89.7 ± 2.8; *Il*CysZ: 87.9 ± 2.5; all values in μmol SO$_4^{2-}$ x mg CysZ$^{-1}$ x min$^{-1}$). Taking into account the actual amount of CysZ incorporated into the proteoliposome preparations (determined with the Amidoblack protein assay, [*Schaffner and Weissmann, 1973*]) revealed catalytic turnover numbers ($k_{cat}$) of 46.4 ± 0.7 s$^{-1}$; 42.5 ± 1.3 s$^{-1}$, and 41.9 ± 1.2 s$^{-1}$ for *Pd*CysZ, *Pf*CysZ, and *Il*CysZ, respectively. Whereas this turnover number is well within the range observed for other secondary transporters (*Jung et al., 1998*; *Malinauskaite et al., 2014*; *Eskandari et al., 1997*; *Quick et al., 2001*), it may also reflect the energetically uncoupled flux of SO$_4^{2-}$ along its

concentration gradient, characteristic for a low-turnover channel phenotype (or passive transport/ uniport, or facilitated diffusion) (*Ashcroft et al., 2009*). In fact, calculations of the $SO_4^{2-}$ accumulation in the *Pd*CysZ-containing proteoliposomes at the peak of the uptake (at 15 s) shown in *Figure 4b* reveal that the internal concentration of $SO_4^{2-}$ is only ~5 fold higher than that of the externally added $SO_4^{2-}$, whereas it is ~3 fold elevated at the plateau (120 s), revealing non-concentrative $SO_4^{2-}$ flux by CysZ that is representative for a channel-like mode of action (see Materials and methods for details). Note that applying the same calculation to the MhsT-mediated $Na^+$-coupled L-tryptophan transport in the identical preparation of MhsT-containing proteoliposomes (*Malinauskaite et al., 2014*) yields an ~150 fold internal concentration excess of the amino acid. Also, it seems worth noting that while the $k_{cat}$ takes into account the total amount of CysZ in the proteoliposomes, experimental limitations preclude the accurate determination of the actual number of functional CysZ molecules in the proteoliposomes, thus resulting in a potentially underestimated value for the true $k_{cat}$.

To distinguish between the active and passive $SO_4^{2-}$ flux phenotypes, we performed uptake studies with CysZ in the reconstituted proteoliposome system in the presence of ionophores that dissipate transmembrane ion gradients that could serve as driving force for $SO_4^{2-}$ uptake, i.e., co-transport or symport. Performing uptake of 1 mM $SO_4^{2-}$ using CysZ-containing proteoliposomes preloaded with 200 mM Tris/Mes, pH 7.5 in assay medium composed of 100 mM Tris/Mes, pH 7.5/100 mM NaCl (to generate a sodium motive force, $\Delta\mu_{Na^+}$) or in 200 mM Tris/Mes, pH 5.5 (to generate a proton motive force, $\Delta\mu_{H^+}$) revealed activities that were virtually indistinguishable from experiments with the same internal and external buffer composition (200 mM Tris/Mes, pH 7.5) or by adding the ionophore gramicidin (causing the dissipation of $\Delta\mu_{Na^+}$ and $\Delta\mu_{H^+}$; *Figure 4* and *Figure 4—figure supplement 1c*). We then tested the effect of the membrane potential on CysZ activity. For these studies, CysZ-containing proteoliposomes were preloaded with and assayed in a Tris/Mes-based buffer system (at pH 7.5) that allowed for a different KCl inside/outside ($K_{in}/K_{out}$) distribution (*Figure 4d*). Potassium flux-generated membrane potentials of different polarity were imposed by adding the potassium ionophore valinomycin, revealing that hyperpolarization of the proteoliposomes (inside-negative membrane potentials) did not markedly increase CysZ-mediated $SO_4^{2-}$ flux into the proteoliposomes. However, depolarization (inside positive) caused about a 40 % increase in the uptake activity of all three CysZ isoforms at a $K_{in}/K_{out}$ distribution of 1:100 (producing a theoretical Nernst membrane potential of +118 mV [*Figure 4d*]).

To further characterize the interaction of CysZ with both $SO_4^{2-}$ and $SO_3^{2-}$, we performed binding experiments with detergent-solubilized and purified CysZ using the scintillation proximity assay (SPA) (*Quick and Javitch, 2007*) as well as micro-scale thermophoresis (MST) (*Seidel et al., 2013*). To determine the concentration of half-maximal sulfate binding ($EC_{50}$), we isotopically diluted $[^{35}S]O_4^{2-}$ with non-labeled $SO_4^{2-}$, obtaining an $EC_{50}$ of 3.51 ± 1.9 mM for *Pd*CysZ, 2.24 ± 0.85 mM for *Pf*CysZ, and 2.29 ± 1.04 mM for *Il*CysZ (*Figure 4e*). Competing binding of $[^{35}S]O_4^{2-}$ with increasing concentrations of $SO_3^{2-}$ revealed that CysZ binds $SO_3^{2-}$ with about 4-fold greater apparent affinity as reflected by a half-maximum inhibition constant ($IC_{50}$) of 0.74 ± 0.23 mM for *Pd*CysZ, 0.60 ± 0.21 mM for *Pf*CysZ, and 0.56 ± 0.20 mM for *Il*CysZ (*Figure 4f*). Whereas the SPA-based binding competition of $[^{35}S]O_4^{2-}$ binding by $SO_3^{2-}$ represents an indirect assay, MST-based binding of $SO_3^{2-}$ revealed apparent affinities for $SO_3^{2-}$ binding by *Pd*CysZ and *Il*CysZ of 3.73 ± 0.76 mM and 8.68 ± 1.88 mM, respectively (*Figure 4—figure supplement 1d*).

Finally, we also employed single-channel electrophysiological current recordings with *Il*CysZ reconstituted in a planar lipid bilayer. Traces often showed multiple levels of a basic conductance, consistent with the incorporation of more than one reconstituted channel (*Figure 4—figure supplement 1e*). Recordings were made with varied membrane potentials applied in symmetrical $Na_2SO_4$ solutions on both sides of the bilayer, from which a single-channel conductance of ~92 pS was deduced. Measurements made with *Pd*CysZ gave a similar single-channel conductance level. Thus, consistent with our $[^{35}S]O_4^{2-}$-based flux assays, these electrophysiological measurements showed that reconstituted CysZ exhibits channel-like properties.

## Sulfate binding site

A bound $SO_4^{2-}$ ion was observed in the structure of *Il*CysZ. This binding site was confirmed by purifying and crystallizing CysZ in the presence of the heavier $SO_4^{2-}$-analog selenate ($SeO_4^{2-}$). In the $SO_4^{2-}$-bound structure, each protomer in the dimer showed electron density consistent with a bound

sulfate, but only one of the two refined to full occupancy, hence only one is shown in the refined model of *Il*CysZ (*Figure 5a* and *Figure 5—figure supplement 1*). The location of the bound $SO_4^{2-}$ is on the outer periphery of the CysZ protomer, close to the putative membrane interface where it is coordinated by two arginine residues (R27 and R28 in *Il*CysZ) and the backbone amides of residues G25 and L26. This $SO_4^{2-}$-binding site is in the loop between helices H1 and H2, with the motif GLR (R) being well conserved among the CysZ family. Consistent with this observation, crystals of *Il*CysZ in which R27 and R28 were replaced with alanines did not show interaction with $SO_4^{2-}$ in that site based on i) the lack of any density in the refined mutant structure at 2.3 Å and ii) functional studies (see below). We could not confirm the presence of $SO_4^{2-}$ in either of the other structures of *Pf*CysZ and *Pd*CysZ, owing, at least in part, to their lower resolution limit (~3.5 Å). Soaking and co-crystallization attempts on both *Pf*CysZ and *Pd*CysZ with $SeO_4^{2-}$ resulted in cracking and destabilization of crystals, and loss of measurable diffraction. In this vein, *Il*CysZ R27A/R28A, and the corresponding

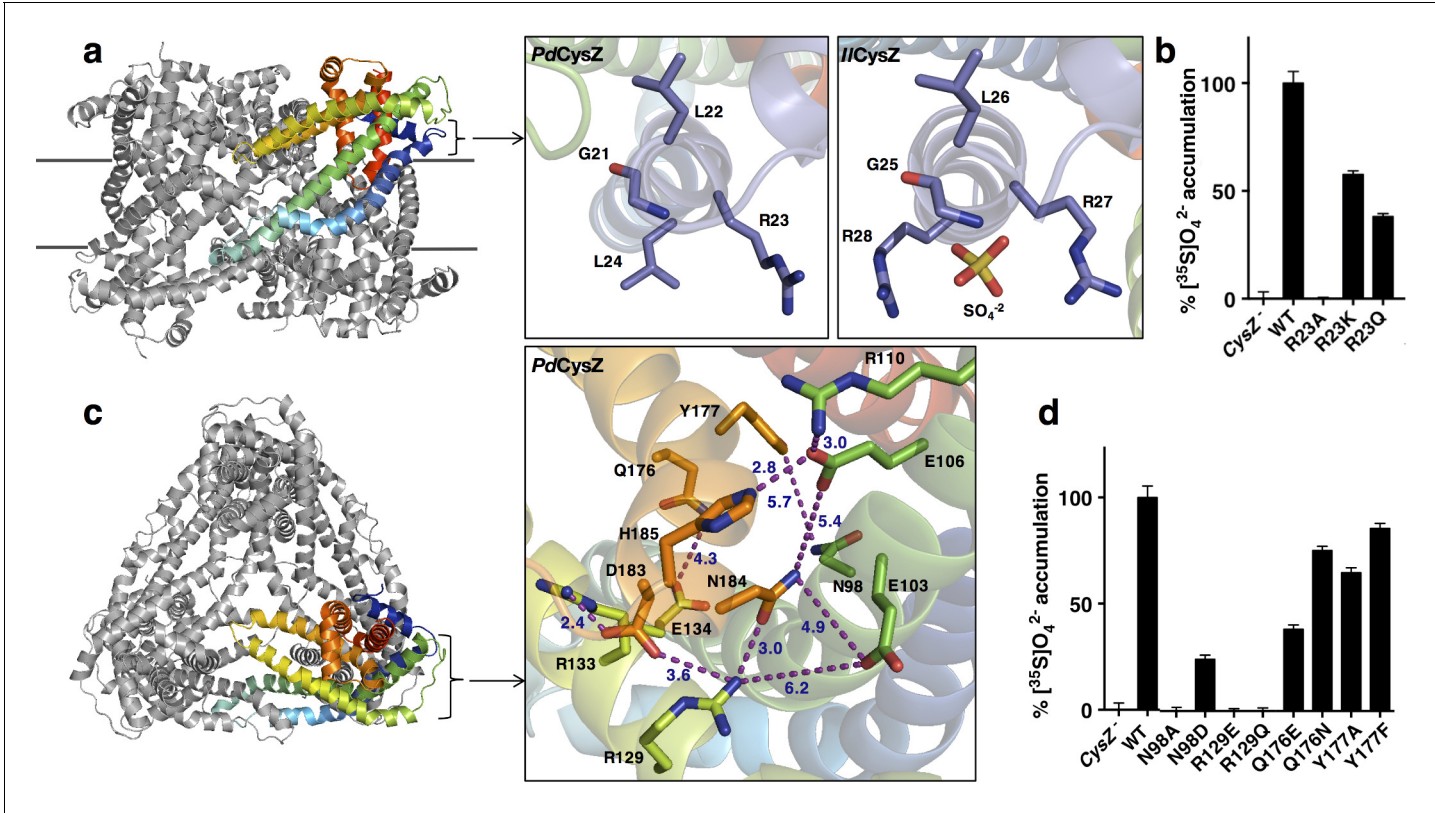

**Figure 5.** Functionally relevant features of the CysZ molecule. (a) Side-view of a ribbon diagram of the *Pd*CysZ hexamer, with one chain rainbow-colored from N- (blue) to C-termini (red). Insets show the sulfate-binding site located at the start of H2a in *Pd*CysZ (left), with conserved residues G21, L22, R23 and L24 labeled, and the corresponding site in *Il*CysZ (right), with residues G25, L26, R27, and R28 labeled and showing the $SO_4^{-2}$ ion as bound in the crystal structure. (b) Sulfate uptake by sulfate-binding site mutants of *Pd*CysZ. *E. coli* K-12 CysZ⁻ cells transformed with the listed *Pd*CysZ R23 mutants were used to measure 320 μM [³⁵S]$O_4^{-2}$ uptake (n = 3). Sulfate uptake was abolished for R23A, and rescued to 50% and 40% of the wild type levels for the R23K and R23Q mutants respectively. (c) A top-view of the *Pd*CysZ hexamer, colored as in a. The inset magnifies the central core of CysZ to show the associated network of hydrogen bonds (R129-N184, E106-H185), van der Waals interactions (E103-N184, E106-N184, Q176-E134) and salt bridges (R129-D183, R133-D183, R110-E106) between pairs of highly conserved residues. Interatomic contacts are shown as purple dotted lines with distances (in Å) marked in blue. (d) Sulfate uptake by central-core mutants of *Pd*CysZ. *E. coli* K-12 CysZ⁻ cells transformed with the listed *Pd*CysZ mutants were used to measure 320 μM [³⁵S]$O_4^{-2}$ uptake (n = 3). N98A and R129E and R129Q showed severely impaired sulfate uptake, whereas more conservative substitutions such as N98D, Q176E and Q176N had less of a negative effect on function. Y177A and Y177F do not show any impaired function.

DOI: https://doi.org/10.7554/eLife.27829.012

The following figure supplement is available for figure 5:

**Figure supplement 1.** Substrate molecules bound at sulfate binding site of *Il*CysZ.

DOI: https://doi.org/10.7554/eLife.27829.013

*Pf*CysZ R25A and *Pd*CysZ R23A mutants all have severely impaired sulfate uptake capability, as demonstrated in cells and proteoliposomes (*Figures 5b* and *Figure 4—figure supplement 1a*), and exhibit dramatically reduced sulfate binding activity (<20% compared of that measured for CysZ-WT, data not shown), consistent with a conserved role for these targeted arginine residues in all CysZ variants.

## Conserved core in the hydrophilic head

Each apex of the triangular faces of the *Pd*CysZ hexamer comprises an extra-membranous hydrophilic head of a CysZ protomer (*Figures 1a* and *5c*). The central core of this hydrophilic head consists of the ends of helices H3 and H5 and the start of helix H4, with their residues forming an intricate network of hydrogen bonds and salt bridges that hold this helical bundle together (*Figure 5c*). Two highly conserved motifs in this region, ExVE and QYxDYPxDNHK (*Figure 2c*), likely play critical roles in this network of interactions. The two aspartates E103 and E106 (*Pd*CysZ) in the first motif interact with R129, N184 and R110, H185, respectively (*Figure 5c*). The conserved R129 and R133 on helix H4b in turn also interact with D183 and N184 in the DNHK sequence stretch of helix H5b in the second motif. In the second motif, a conserved tyrosine (Y177) lies below the membrane interface towards the core of the molecule, along helix H5.

Mutational studies on a subset of these conserved hydrophilic-head residues highlight their functional relevance (*Figure 5d*). R129 mutants (R129E or Q) exhibited a severe loss of $SO_4^{2-}$ binding and uptake, and mutations made to Q176 and Y177 also showed loss in function, albeit to a lesser extent. Charge reversal mutations made to E103 and E106 (to K or R) or the equivalent *Il*CysZ E107, E110 resulted in very poor expression and stability levels, suggesting that the disruption of the interaction network in this region may destroy the structural integrity of the protein.

## Putative pore and sulfate translocation pathway of CysZ

It was already evident from the initial *Il*CysZ structure that the hydrophilic head presented an incipient opening into the membrane (*Figure 5c*-inset, *Figure 6a*), but it was quite unclear how this opening might relate to transmembrane sulfate translocation. The 'transverse' dimer structure of *Pf*CysZ clarified the possibility for ion permeation by showing that openings in its two protomers aligned across the putative membrane; however, the prospective transduction pathway would then be open on one side to the bilayer (*Figure 6—figure supplement 1*). The *Pd*CysZ structure showed that the open sides of transverse dimers line a central cavity in the *Pd*CysZ hexamers. Thus, plausible transduction pathways became more evident.

The view into a protomer surface along the direct pathway between transverse dimer apices (e.g. green to orange in *Figure 3e*) displays the pattern of exceptionally high conservation associated with the entrance to a putative pore for sulfate translocation (*Figure 6a,b*). This putative entrance or 'pore' lies in the midst of a tight network of interacting residues, which is seen detailed in the inset of *Figure 5c* in the very same view as for *Figure 6a*. These interactions close the incipient pore from ion conduction in this conformation. Intriguingly, this putative pore-entrance lies approximately 17–20 Å (linear distance) away from the observed sulfate binding site. The electrostatic potential surface of PdCysZ (*Figure 6c*) shows striking and puzzling electronegativity at the conserved pore entrance (*Figure 6d* compared with *Figure 6b*). Because of the conservations, similar electrostatics pertain to the two other homologs. Moreover, we observe the same helix dispositions, pore shape and charge interactions in all three structures, evident when the protomers are superimposed (*Figure 3c*, top view).

Pathway prediction algorithms (namely, PoreWalker [*Pellegrini-Calace et al., 2009*]) performed on the CysZ protomer revealed a putative ion translocation pathway that begins at the entrance to the putative pore and ends in the large central cavity of the CysZ hexamer that lies within the membrane (*Figure 7a,b*). In the structure of *Pd*CysZ, the entrance of the putative pore appears to be closed by the network of charge-interactions by conserved residues, R110, E106, N184, H185 and W235 (*Figures 6a and 7c–1*). These residues interact to tightly restrict access to the putative translocation pathway and constrict the entry of ions, likely selected for size and charge (*Figure 7—figure supplement 1a*). After the narrow entrance, the putative pathway broadens, surrounded by a ring of conserved asparagine and glutamine residues, N33, N91, N98, Q176, and N220 (*Figure 7c-2*). Mutations to N98 (N98A, N98D) led to impaired sulfate uptake (*Figure 5d*). Following this polar

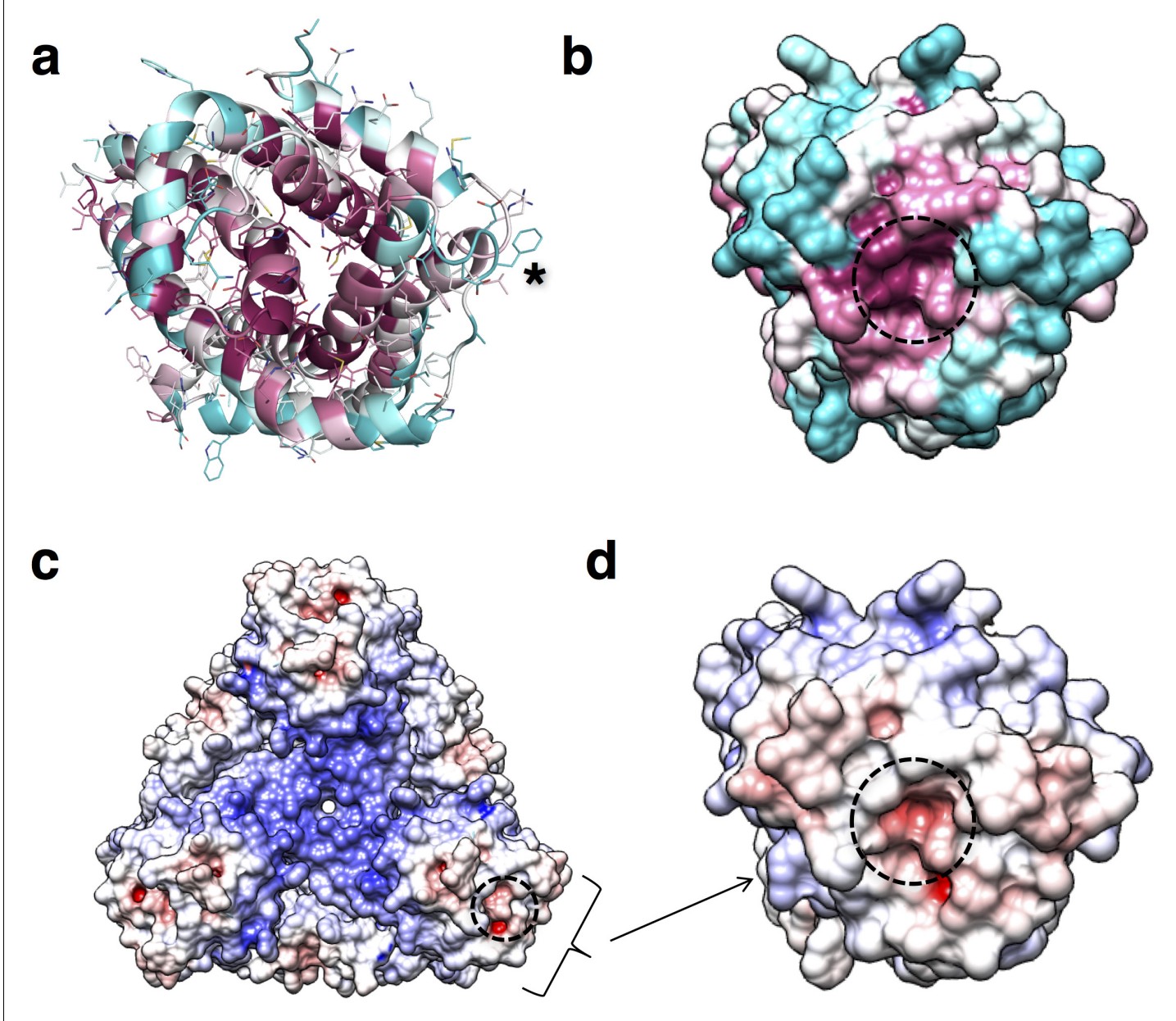

**Figure 6.** Conservation and entrance of putative pore of *Pd*CysZ. (a) Ribbon diagram colored by conservation with residues in maroon being most conserved to cyan being least conserved (calculated by ConSurf) to highlight the entrance to the putative pore; an asterisk (*) marks the location of the sulfate-binding site (GLR motif) at top of helix H2a. (b) Same view and coloring scheme as in a, but now shown in surface representation. (c) Electrostatic representation of hexameric *Pd*CysZ as viewed from the top, with negative surface potential represented in red, and positive potential in blue as calculated by APBS, with location of the putative pore marked by a dashed circle. (d) Close-up view in electrostatic representation of the putative pore within a *Pd*CysZ protomer, surface and orientation as in b.

DOI: https://doi.org/10.7554/eLife.27829.014

The following figure supplement is available for figure 6:

**Figure supplement 1.** Visualization of putative pore of *Pf*CysZ.
DOI: https://doi.org/10.7554/eLife.27829.015

environment, the putative pathway widens even further leading into a large, primarily hydrophobic, internal cavity encapsulated by the TM helices of the *Pd*CysZ hexamer, located in the plane of the lipid bilayer (*Figure 7b* and *Figure 7—figure supplement 1c,d*). The hydrophobic cavity is enclosed on both the top and bottom by pairs of helices H4-H5, three on each side from the six protomers,

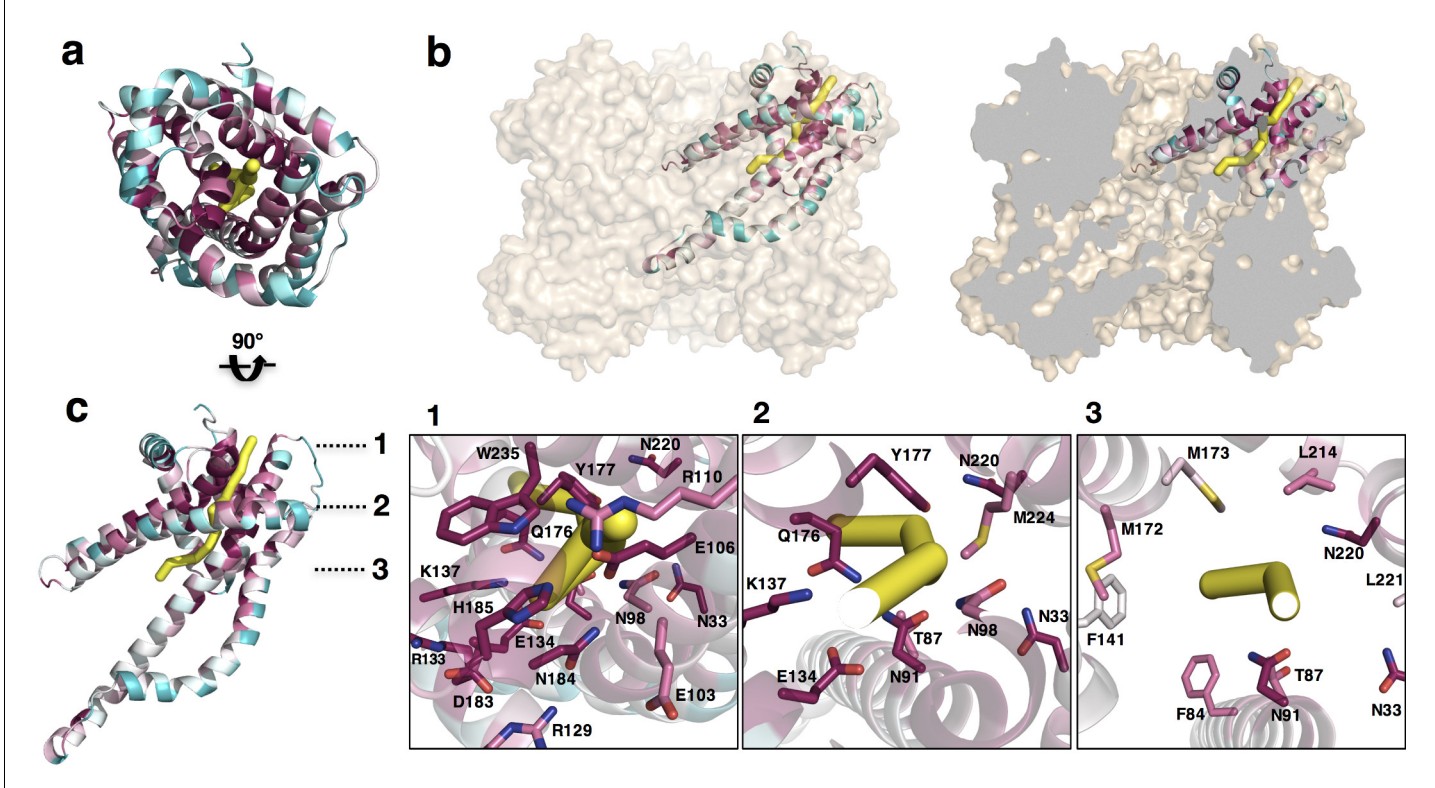

**Figure 7.** Putative ion conductance pathway of CysZ. (a) A *Pd*CysZ protomer looking into the incipient pore entrance. The polypeptide ribbon is oriented as in *Figure 6a* and colored by level of sequence conservation (calculated by ConSurf, with maroon being most conserved and cyan least). The putative pore and ion conduction pathway is shown as a yellow tube of 1 Å diameter, calculated by PoreWalker. (b) Surface representation of the *Pd*CysZ hexamer, as side views, both semi-transparent (left) and with surface clipped (right, cut surfaces colored in grey) to allow for the internal visualization of the pathway leading to the central cavity. (c) Side view of a *Pd*CysZ protomer left, viewed as in b and rotated 90° from a. Insets show magnified views of cross-sections along the pathway: At level 1, the entrance to the pore consists of a narrow constriction created by a network of highly conserved polar and charged residues (E106, R110, E134, N184, H185) tightly interacting with one another. At level 2, the pathway broadens and becomes less charged, but yet polar in nature as lined by conserved asparagine, tyrosine and threonine residues (N33, T87, N91, N98, Q176, Y177). From level 3, the pathway widens further and ultimately leads into the large central hydrophobic cavity.

DOI: https://doi.org/10.7554/eLife.27829.016

The following figure supplement is available for figure 7:

**Figure supplement 1.** Plot of the pore diameter of the putative ion conduction pathway.

DOI: https://doi.org/10.7554/eLife.27829.017

pointing in towards the three-fold axis, with each side of the cavity, at its mid-section, measuring ~50 Å, when viewed from above or outside the membrane (*Figure 7—figure supplement 1c,d*). Given the symmetric, dual topology nature of the CysZ assembly, the ions could then exit the cavity via the same pathway as they entered, but traveling through CysZ protomers located on the opposite side of the membrane.

## Discussion

There are four known families of dedicated $SO_4^{2-}$ transport systems in prokaryotes[6], of which CysZs are the least studied (*Zhang et al., 2014*). The sequence of CysZ, coding for an integral membrane protein with four predicted TM segments, shows no resemblance to any other known protein. This, and the quest to set the basis for a mechanistic understanding of function, prompted us to investigate the structure of CysZ.

We present here three structures of CysZ from different species, all determined by x-ray crystallography. These structures are all similar, showing a novel fold comprising two extended TM helices and two hemi-penetrating helical hairpins, giving rise to a tripod-like shape within the membrane,

and a hydrophilic head (*Figures 2a,b* and *3c*). Two of the structures (*Il*CysZ and *Pf*CysZ) show dimers in the crystals, albeit with different interfaces (*Figure 3a,b*), and the third structure (*Pd*CysZ) displays a hexameric assembly in multiple crystal forms (*Figure 1a*).

All three structures have a dimeric component, each of which is present in the hexameric arrangement present in all the crystal forms of *Pd*CysZ (*Figure 3a,b*), showing how they are all related. Furthermore, *Il*CysZ can be crosslinked in membranes to the alternative *Pf*CysZ dimer by engineering of disulfide cysteine mutants (*Figure 3—figure supplement 2*), which suggests that *Il*CysZ can adopt both of the dimer assemblies observed in the *Pd*CysZ hexamer. Thus, we hypothesize that CysZ is a hexamer in nature, as observed in *Pd*CysZ, consistent with the fact that all CysZs studied here exhibit essentially the same functional properties. In the case of *Il*CysZ and *Pf*CysZ, this assembly may have come apart into the subsequently crystallized dimeric components during the process of detergent extraction from the membrane and purification. This could be explained by the unusually labile, and conformationally-flexible, nature of the H4b-H5a helical hairpin of CysZ, with only one pair of stabilizing TM helices per protomer (H2-H3). We presume that the hexamer is maintained with the support and scaffolding of the lipid bilayer.

CysZ shows an inverted transmembrane arrangement, which we confirmed with cross-linking experiments on membranes (*Figure 3—figure supplement 2*). Antiparallel insertion is fairly uncommon in membrane proteins; however, reported cases include the well-documented EmrE, a multi-drug resistant export protein that inserts into the membrane as an anti-parallel dimer, and the more recently discovered and studied family of double-barreled 'Fluc' fluoride channels (*Amadi et al., 2010*; *Korkhov and Tate, 2009*; *Rapp et al., 2006*; *Stockbridge et al., 2015*; *Stockbridge et al., 2013*).

Functional characterizations of the three CysZs for which we have obtained structural information show that all three mediate $SO_4^{2-}$ flux and that this flux is inhibited by $SO_3^{2-}$, thus supporting previously reported data on *E. coli* CysZ (*Zhang et al., 2014*). However, in contrast to what was observed for *E. coli* CysZ (*Zhang et al., 2014*), our data indicate that $SO_4^{2-}$ flux by *Pd*CysZ, *Pf*CysZ and *Il*CysZ is not thermodynamically coupled to the proton gradient (i.e., $H^+/SO_4^{2-}$ symport) or the $Na^+$ gradient (i.e., $Na^+/SO_4^{2-}$ symport) but rather reflect the non-concentrative flux of $SO_4^{2-}$ characteristic of a passive, channel-like translocation mechanism. However, $SO_4^{2-}$ flux in CysZ-containing proteoliposomes differed from the ohmic *I-V* relationship indicative of a single ion-selective channel in that it was not affected by hyperpolarization (inside negative), but depolarization of the proteoliposomes (inside positive) led to an increased inward flux of $SO_4^{2-}$. Thus, we contemplate that CysZ $SO_4^{2-}$ flux activity may involve co-permeation of other ions with $Na^+$ being the most likely for our experimental conditions. Our electrophysiological measurements of purified CysZ incorporated into a lipid bilayer did not aim to assess permeability ratios of $SO_4^{2-}$ and other ions, i.e., for $Na^+$, symmetric $Na_2SO_4$ solutions were used in the experiments.

Building on our functional and structural discoveries, our results suggest a fascinating hypothesis for mechanisms of $SO_4^{2-}$ transfer and regulation. There is a conserved central core (*Figure 6a*), which is likely to have a structural role. Close to this lies a sulfate-binding site, with functional implications (*Figure 5a*), and the entrance to a putative pore (*Figure 5c*). The entrance of this putative pore is delineated by a hydrophilic network of conserved residues that form a tight constriction in our observed conformation. Following this hypothetical route, sulfate ions that might enter through the three separate pores, one per CysZ dimer pair, would converge into a central hydrophobic cavity (*Figure 7b* and *Figure 7—figure supplement 1c,d*). Once sulfate ions enter this central cavity, due to the unfavorable environment, these are likely to exit it rather rapidly through one of the three available exit pores on the cytoplasmic side of the hexamer. Hydrophobic cavities and pores are seen commonly in ion channels, with examples ranging from the well documented hydrophobic inner pores of the various potassium channels (*Doyle et al., 1998*) to the SLAC1 (*Chen et al., 2010*) and bestrophin anion channels (*Yang et al., 2014*), and the MscS and MscL mechanosensitive channels (*Anishkin et al., 2010*; *Bass et al., 2002*; *Birkner et al., 2012*; *Chen et al., 2010*; *Doyle et al., 1998*), facilitating the rapid passage of ions due to the unfavorable environment, as well as potentially providing a means of 'hydrophobic gating' (*Aryal et al., 2015*).

The surface electrostatics of CysZ (*Figures 1b* and *6c,d*) draw attention to the negative potential of the conserved core of each protomer, which is then surrounded by a more neutral annulus. While this feature seems contradictory to admission of $SO_4^{2-}$ ions at the extracellular side, it could be advantageous for expulsion into the cytoplasm on the opposite side. In any case, the structures that

we have determined are evidently in a closed state, implying that a conformational change would have to occur to allow passage of $SO_4^{2-}$, likely modifying the surface electrostatics of the protein. It is tempting to speculate that the regulation of the opening of the pore could be modulated by the binding of sulfate ions to the identified sulfate-binding site, as it is near the entrance of the putative pore. L22 lies in proximity of the entrance of the putative pore, and its backbone amide (along with G21) coordinates the sulfate ion in the GLR motif of the sulfate-binding site. Thus, the binding of sulfate to the GLR motif could trigger a conformational change needed to displace L22, allowing for a wider opening for the sulfate ions to enter the pore. Sulfite could hypothetically exert its inhibitory effect on CysZ function by binding to this site.

There is an overall electropositive region in the center of the hexameric molecule, when viewed from the top (*Figures 1b* and *6c,d*). This central region is lined by conserved residues along helices H4b-H5a, namely, R129, R133 and K137. The hydrophobic tips of helices H4b-H5a of the three protomers on each side of the membrane then converge in the center of the hexamer. A pore through the three-fold axis of the hexamer could provide an alternative passageway for $SO_4^{2-}$ ions through this assembly.

We observe, in agreement with previous data, that $SO_3^{2-}$ inhibits CysZ-mediated $SO_4^{2-}$ flux (*Figure 4b,d* and *Figure 4—figure supplement 1a,f*) (*Zhang et al., 2014*). The antiparallel nature of CysZ could provide a means for internal as well as external regulation. However, experiments in solutions – such as crystallizations – where all molecules are exposed to the same chemical environment, makes capturing such a state in an open or $SO_3^{2-}$ blocked conformation challenging. Despite this limitation, our CysZ structures and associated functional experiments have allowed us to make substantial progress in the understanding $SO_4^{2-}$ uptake by these membrane permeases. This work sets the framework for future experiments aimed at unraveling the molecular details of how $SO_4^{2-}$ is translocated across the membrane by CysZ and how this process is regulated.

## Materials and methods

### Ortholog selection and cloning

A total of 63 *cysZ* candidate genes were selected by a bioinformatics approach implemented by the New York Consortium of Membrane Protein Structure (NYCOMPS), as previously described (*Punta et al., 2009*). The majority of the genes (including *Il*CysZ, uniprot ID: Q5QUJ8) were PCR-amplified from fully sequenced prokaryotic genomic DNA (obtained from ATCC) (*Love et al., 2010*). CysZ genes from certain species, such as *Pf*CysZ (uniprot ID: A0A0X8F058) and *Pd*CysZ (uniprot ID: M4XKU7) were chemically synthesized by GenScript (Genscript, Piscataway, New Jersey), with codon-optimization for protein expression. All genes were cloned by ligation-independent cloning (LIC) (*Aslanidis and de Jong, 1990*) into an IPTG (isopropyl β-D-1-thiogalactopyranoside) inducible, kanamycin-resistant pET derived plasmid (Novagen, Madison, Wisconsin), with an N-terminal deca-histidine tag (His10) and a TEV (tobacco etch virus) protease site to allow for tag cleavage upon purification.

### Protein expression and purification

Expression plasmids bearing the *cysZ* genes were transformed into BL21(DE3)pLysS cells using standard protocols, and grown at 37°C in 2XYT media supplemented with 50 µg/ml kanamycin and 50 µg/ml chloramphenicol in an orbital shaker at 250 rpm. Protein expression was induced for ~16 hr at 22°C with 0.2 mM IPTG once an absorbance ($A_{600\ nm}$) of 0.8–1.0 was reached. Selenomethione (Se-Met)-incorporated proteins were expressed in BL21(DE3)pLysS cells grown using an M9 minimal media kit (Shanghai Medicilon, China) supplemented with the necessary minerals, vitamins and non-inhibitory amino acids. Se-Met was added prior to IPTG induction at an $A_{600\ nm}$ of 1.2. The Se-Met-incorporated protein was purified using the same procedures as the native protein. Once harvested, the cells were resuspended at 0.2 g/ml in lysis buffer containing 20 mM Na-Hepes pH 7.5, 200 mM NaCl, 20 mM MgSO4, DNase I and RNase A, 0.5 mM PMSF (phenylmethylsulfonyl fluoride), EDTA-free Complete protease inhibitor cocktail (Roche, Switzerland) and 1 mM TCEP-HCl (Tris (2-carbox-yethyl) phosphine hydrochloride) as a reducing agent. Initial small-scale expression and detergent screening was performed on 80 mg of pelleted cells (wet weight), and 7–10 g of cells for large-scale protein purification. Cells were lysed using an Avestin® EmusiFlex-C3 homogenizer, followed by

protein solubilization with 1% (w/v) decyl maltopyranoside (DM) (Anatrace, Maumee, Ohio) for 1 hr at 4°C, after-which insoluble material was removed by ultra-centrifugation at 100,000 x g. The solubilized protein was applied to Ni-NTA Sepharose (Qiagen, Germantown, Maryland) in batch, washed with lysis buffer containing 0.2% DM and 40 mM imidazole and eluted in buffer containing 250 mM imidazole. Upon elution, CysZ was dialyzed overnight with His-tagged TEV protease at 4°C, against a buffer containing 20 mM Na-Hepes pH 7.0, 200 mM NaCl, 0.2% DM, 1 mM TCEP-HCl and 20 mM $Na_2SO_4$, allowing for the cleavage of the His10 tag and removal of the imidazole. Tagless CysZ was then re-passaged over Ni-NTA sepharose to re-bind of any uncleaved CysZ, TEV protease and the cleaved His10 tag. The protein was then subjected to size-exclusion chromatography (Superdex 200 10/30 HR; GE Healthcare, Chicago, Illinois) in 20 mM Na-Hepes pH 7.0, 200 mM NaCl, 1 mM TCEP-HCl, 20 mM $Na_2SO_4$ and appropriate detergent for crystallization, for *Il*CysZ: 0.06% Lauryl dimethylamine oxide (LDAO) and for *Pf*CysZ and *Pd*CysZ: 1% β-octyl glucopyranoside (β-OG). The choice of detergent was made based on protein yield, stability and mono-disperse gel-filtration peaks obtained in the initial small-scale detergent screening. A yield of ~1.5 mg of purified CysZ was typically obtained from a cell pellet of 7–8 grams (1 liter of culture).

## Protein crystallization

### *Il*CysZ
Crystals of *Il*CysZ in LDAO were obtained by vapor diffusion at a protein concentration of 6–8 mg/ml at 4°C, in a 1:1 v/v ratio against a precipitant of 28–32% PEG400, 0.1M Tris-HCl pH 8.0, with salt additive of 0.1 M NaCl or 0.1 M $MgCl_2$. The crystals appeared overnight, continued to grow in size over the course of 2–4 days after set-up. After optimization, the crystals grew to a maximum size of ~200 μm x 100 μm x 50 μm with a rhomboid or cuboid shape. The crystals were harvested directly without the addition of a cryo-protectant and flash-frozen into liquid nitrogen, for data collection on the X4A/X4C beamlines at National Synchrotron Light Source (NSLS), Brookhaven National Labs (Upton, NY). SeMet derivatized and selenate co-crystals were obtained from the same conditions as the native crystals.

### *Pf*CysZ
Crystals of *Pf*CysZ in β-OG at 5 mg/ml were initially obtained at 4°C by vapor diffusion, in a 1:1 protein to precipitant ratio, after 1–2 days against 28% PEG400, 0.1 M MES pH 6.0. They were cuboid in shape and grew in clusters of multiple crystals originating from a common locus. The crystals were optimized to a maximal size of ~150–200 μm x 50 μm x 50 μm, with the best diffracting crystals grown under silicone oil (visc. 500) in microbatch Terazaki plates. Crystals were directly flash-frozen into liquid nitrogen without the use of a cryo-protectant and were exposed to X-rays at the NE-CAT (24-IDC and IDE) beamlines at APS, Argonne National Lab (Argonne, IL) for data collection. SeMet crystals were obtained from the same conditions as the native protein.

### *Pd*CysZ
Crystals of *Pd*CysZ in β-OG at 5–8 mg/ml were initially obtained at 4°C by vapor diffusion, in a 1:1 protein to precipitant ratio, after 2–3 days against 22–30% PEG550MME, 0.1 M Na-Hepes pH 7.0. The rod-like crystals were hexagonal on one face, and grew in clusters originating from a common locus as well as on the edge of the drop. The multiple crystal forms observed were all obtained in the same crystallization conditions. The crystals were optimized to a maximal size of ~200–250 μm x 25 μm x 50 μm. With the addition of 20% glycerol (w/v) as a cryo-protectant, the crystals were flash-frozen into liquid nitrogen and were exposed to X-rays at the NE-CAT (24-IDC and 24-IDE) beamlines at APS, Argonne National Lab (Argonne, IL) for data collection.

## Data collection and structure determination

### *Il*CysZ
The structure of CysZ was determined by the single-wavelength anomalous diffraction (SAD) method from anomalous diffraction of a selenate ($SeO_4^{2-}$) derivative crystal. The anomalous signals were measured at the Se K-edge peak wavelength, which was determined experimentally from fluorescence scanning of the crystal prior to data collection. All diffraction data were recorded at 100K using an ADSC Q4R CCD detector at the NSLS X4 beamline. Diffraction data were indexed,

integrated, scaled, and merged by HKL2000 (*Otwinowski and Minor, 1997*). Selenate substructure determination was performed with the SHELXD program through HKL2MAP (*Pape and Schneider, 2004*). A resolution cut-off at 2.6 Å was used for finding Se sites by SHELXD. A strong peak found by SHELXD was used to calculate initial SAD phases, which were improved by density modification by SHELXE (*Sheldrick, 2010*). With a solvent content of 65% corresponding to two molecules in the asymmetric unit, 50 cycles of density modification resulted in an electron density map of sufficient quality for model building. The initial polypeptide chain was built by Arp/Warp (*Langer et al., 2008*), at 2.1 Å by using experimental phases. Further cycles of model building were performed manually using COOT (*Emsley et al., 2010*) and all rounds of refinements were performed with PHE-NIX (*Adams et al., 2010*). The native structure of CysZ with bound sulfate was determined both by multi-crystal native SAD (*Liu et al., 2012*) (final resolution of 2.3 Å) and by molecular replacement with the selenate bound model (final resolution of 2.1 Å). In addition, phase information obtained from Se-Met derivatized protein with 9 Se sites per CysZ molecule, verified our model obtained from the selenate data.

### *Pf*CysZ

Multi-crystal SeMet-SAD data sets were collected at APS beamline 24-IDC with a Pilatus 6M pixel array detector under a cryogenic temperature of 100 K. To enhance anomalous signals from Se atoms for phasing (*Liu et al., 2011*), the X-ray wavelength was tuned to the Se-K edge ($\lambda$ = 0.9789 Å). The orientation of crystals was random without special consideration of crystal alignment, and beam size was adjusted to match the crystal size. A total of 22 data sets were collected, each from a single crystal. An oscillation angle of 1° was used for data collection with a total of 360 frames for each data set. The beam size was adjusted to match the crystal size. The 22 single-crystal data sets were processed individually by using XDS (*Kabsch, 2010*) and CCP4 packages (*Winn et al., 2011*). For phasing purposes, the low-resolution anomalous signals were enhanced by increased multiplicity. By rejection of 7 outlier crystals (*Liu et al., 2012*), anomalous diffraction data from 15 statistically-compatible crystals were scaled and merged for phasing. For outlier rejection, a unit-cell variation of 1.0σ was used. CCP4 program POINTLESS and SCALA (*Evans, 2006*) were used for data combining; and Bijvoet pairs were kept separately throughout the data flow. For refinement purposes, keeping the high angle data was important, and done by limiting radiation damage as well as by increasing multiplicity. Although most *Pf*CysZ crystals diffracted to only about 3.5 Å spacings or poorer, we intentionally set the detector distance to include higher spacings. Higher resolution data were retained through a data merging procedure that is described as follows: (1) The 22 individually processed data sets were analyzed by diffraction dissimilarity analysis by using only high angle data between 3.5 and 3.0 Å, resulting in three subsets. (2) The data statistics of members in each subset were checked manually and the subset that contained the highest angle data set, e.g. data set 6, was selected for further analyses and data combination. (3) Each data set within the selected subset was compared with data set six by high-angle intensity correlation. Six of the highest resolution data sets were statistically comparable and therefore were selected for merging. For phasing, substructure solutions were found by SHELXD (*Sheldrick, 2010*) and were further refined and completed by PHASER (*McCoy et al., 2007*) and then used to compute initial SAD phases at the data limit by SAD phasing with PHENIX (*Adams et al., 2010*). Phases were density modified with solvent flattening and histogram matching as implemented in CCP4 program DM (*Cowtan and Zhang, 1999*) to improve phases and also to break phase ambiguity. The estimated solvent contents of 71% were used for density modification. The model was initially built into the experimental electron density map by COOT (*Emsley et al., 2010*), followed by iterative refinement by PHENIX and model building in COOT. The refined model does not contain solvent molecules at this resolution.

### *Pd*CysZ

Native crystal data were collected at the APS beamline 24-IDC with a Pilatus 6M pixel array detector at a cryogenic temperature of 100 K at an X-ray wavelength of $\lambda$ = 1.023 Å. The sample-to-detector distance was set to 500 mm. An oscillation angle of 0.5° was used for data collection. The beam size was adjusted to match the crystal size. Molecular replacement was attempted using a variety of search models (*Pf*CysZ and *Il*CysZ monomer/dimer models, with various degrees of truncation). Success was achieved by searching for six copies of a search model consisting of the *Pf*CysZ monomer,

with residues 36–53 deleted and the sequence adjusted using CHAINSAW (*Stein, 2008*) (pruning non-conserved residues to the gamma carbon). Density modification of the initial map was performed in PARROT, incorporating solvent flattening, histogram matching and NCS-averaging. An initial round of model building was performed in COOT (*Emsley et al., 2010*) into this map, followed by further phase improvement and bias-removal using *phenix.prime_and_switch* (*Adams et al., 2010*). The improved map was used for a second round of model building, followed by iterative cycles of reciprocal space refinement using *phenix.refine*, and real-space refinement and correction in COOT.

All graphical representations and figures of our structural models were made in either PyMOL (*Schrodinger, 2010*) or Chimera (*Pettersen et al., 2004*).

## $[^{35}S]O_4^{2-}$ Uptake Experiments

### In whole cells

The wild-type parental *E. coli* K-12 strain BW25113 (F⁻, Δ(araD-araB)567, ΔlacZ4787(::rrnB-3), λ⁻, rph-1, Δ(rhaD⁻ rhaB)568, hsdR514) and the *E. coli* K-12 *cysZ* knockout strain JW2406-1 (F⁻, Δ(araD-araB) 567, ΔlacZ4787(::rrnB-3), λ⁻, ΔcysZ742::kan,rph-1, Δ(rhaD⁻ rhaB)568, hsdR514) were obtained from the Coli Genetic Stock Center at Yale University (http://cgsc.biology.yale.edu/), originally found in the Keio knockout collection (*Baba et al., 2006*). Cells were made competent by standard protocols (*Hanahan, 1983*). The strains were grown in LB (Luria Broth) without any antibiotic (BW25113) or with 50 µg/ml of kanamycin (JW2406-1) at 37°C overnight to stationary phase and the cultures were collected by centrifugation at 3000 x g. Cells were resuspended in Davis-Mingioli (DM) minimal media without sulfate (MgSO$_4$ was replaced with MgCl$_2$ and (NH$_4$)$_2$SO$_4$ was replaced by NH$_4$Cl), supplemented with 0.63 mM L-cysteine and were allowed to grow in the absence of sulfate (*Davis and Mingioli, 1950*) for a minimum of 9 hr but no longer than 16 hr at 37°C to ensure that the cells were starved of sulfate and the sulfate stores in the cell were depleted to enhance sulfate uptake measurements (F. Parra, personal communication). The cells were collected, washed 3 times in 5 mM Na-Hepes pH 7.0, and resuspended in the same buffer at 0.7 mg cell protein/ml at room temperature. Uptake (performed in triplicate) was initiated by the addition of 320 µM of Na$_2$[$^{35}$S]O$_4$ to the cell suspension (at a final cell protein concentration of 0.07 mg/mL) and was measured over a time course of typically 0–300 s. The reaction was stopped by diluting the reaction mixture with 1.5 ml of ice-cold 5 mM Na-Hepes, pH 7.0 and immediately filtered through glass-fiber filters (0.75 µm, GF/F). Filters were washed once more with 1.5 ml of buffer, dried, and incubated with EconoSafe scintillation cocktail for >12 hr prior to counting of the radioactivity in a Hidex 300 SL scintillation counter. For the *CysZ*⁻ cell rescue experiments, a similar protocol was used, transformed with an ampicillin resistant expression vector containing the *cysZ* gene, mutant or empty vector (as the control). Upon transformation, the cells were grown overnight in a 4 ml starter culture of LB with 50 µg/ml kanamycin and 100 µg/ml ampicillin. The next morning the entire 4 ml was used to inoculate 75 ml of LB (Kan, Amp), and grown at 37°C for 2.5–3 hr until an OD$_{600}$ of 0.6–0.8 was reached. Protein expression was then induced at 37°C for 4 hr with 0.2 mM IPTG. After 4 hr, the cells were spun down and resuspended in the DM minimal media without sulfate, 0.63 mM cysteine, Amp, Kan and 0.2 mM IPTG, and incubated overnight at 22°C. The next morning the cells were spun down, washed 3 times with 5 mM Na-Hepes 7.0 and resuspended at final concentration of 0.7 mg/ml for uptake.

### In proteoliposomes

CysZ was purified by the procedure described above, and upon elution from the size-exclusion column concentrated to 1 mg/ml for reconstitution into liposomes at a protein to lipid ratio of 1:100. The liposomes were comprised of a 3:1 ratio of *E. coli* polar lipids and phosphatidylcholine (PC) (Avanti Polar Lipids, Alabastar, Alabama), prepared by previously described methods (*Rigaud et al., 1995*). Typically, 4 µl of CysZ-proteoliposomes (corresponding to ~0.2 µg CysZ) and control 'empty' liposomes at a concentration of 5 mg/ml, were assayed in 100 µl of reaction buffer containing [$^{35}$S] O$_4^{2-}$ (the specific radioactivity was adjusted with Na$_2$SO$_4$) and additions as indicated at the indicated concentrations for the indicated periods of time. Reactions were stopped by the addition of ice-cold 5 mM Na-Hepes, pH 7.0 and filtered through 0.22 µm nitrocellulose filters as described above for whole-cell uptake studies.

## Calculation of internal substrate concentration in proteoliposomes

The determination of lipid molecules in a proteoliposome was performed according to the following calculation: $N_{totPL} = \frac{(4\pi r^2) + (4\pi [r-m]^2)}{a}$ - 8, where $N_{totPL}$ is the total number of lipid molecules per proteoliposome, $r$ the radius (external; 50 nm in a 100-nm proteoliposome), $m$ the membrane thickness (4 nm), $r$-$m$ the radius (internal), and $a$ is the area of the lipid headgroups (0.7 nm$^2$) (Lind, 2015). Accordingly, $N_{totPL}$ = 82866.24. With an average $M_W$ of 790.85 of the lipids in the liposome preparation, the lipid concentration in the assay (4 μL of 5 mg/mL proteoliposomes in 100 μL assay volume) is 0.2 g lipid/L or 2.53 x 10$^{-4}$ M. The number of lipids ($N_{lipids}$) per liter is the product of the lipid concentration and Avogadro's number ($N_A$) to be 2.53 x 10$^{-4}$ mol/L x 6.022 x 10$^{23}$ mol$^{-1}$ = 1.523 x 10$^{23}$ lipids/L. The number of proteoliposomes in the sample ($N_{PL}$) can be calculated according to $N_{lipids}$/$N_{totPL}$, yielding $N_{PL}$ = 1.523 x 10$^{23}$ lipids/L x 82,866.24 lipids/proteoliposome = 1.838 x 10$^{15}$ proteoliposomes/L or 1.838 x 10$^{11}$ proteoliposomes/100 μL. The intraliposomal volume ($V_{PL}$) in a 100 μL-assay is calculated according to $V_{PL} = \left(\frac{4}{3}\pi [r-m]^3\right) x\ N_{PL}$ - 8 = 7.49 x 10$^{-8}$ L or 0.0749 μL. In Fig. 4b, SO$_4$$^{2-}$ accumulation at the 15-s time point for $Pd$CysZ is 0.1869 nmol, yielding a concentration of 2.49 mM SO$_4$$^{2-}$; the SO$_4$$^{2-}$ accumulation at the plateau (120-s time point) is 0.1145 nmol, yielding an internal concentration of 1.53 mM. Thus, at an external SO$_4$$^{2-}$ concentration of 0.5 mM, this is a ~ 5-fold concentration excess at the peak uptake point (15 s) or a ~ 3-fold excess at the plateau level. For comparison of the CysZ-mediated SO$_4$$^{2-}$ accumulation to the concentrative transport of a Na$^+$-coupled secondary transporter, we performed the same calculation for $^3$H-Trp transport in proteoliposomes containing the multi-hydrophobice substrate transporter, MhsT (Malinauskaite, 2014). The turnover of Trp transport was determined to be 0.8±0.03 s$^{-1}$. The accumulation of 2.5 μM Trp (about $K_m$) at the 10-s time point was 2.78 x 10$^{-11}$ mol, yielding an internal concentration of the amino acid of 373.8 μM – a ~150-fold excess of the accumulated amino acid vs the external concentration of 2.5 μM.

## Radioligand binding by Scintillation Proximity Assay (SPA)

CysZ was purified by standard protocols as described above, with the exception of leaving the histidine-tag intact without cleavage by the TEV protease. The imidazole was removed by dialysis and the purified protein was not run over the size exclusion column, and instead was directly concentrated to 2 mg/ml for the SPA experiment. [$^{35}$S]O$_4$$^{2-}$ obtained in the form of sulfuric acid (American Radiolabeled Chemicals (ARC), St. Louis, Missouri) was used as the radioligand. 200 ng of CysZ was used per assay point, diluted in 100 μl of assay buffer containing 20 mM Hepes pH 7.0, 200 mM NaCl, 0.2% DeM, 20% glycerol, 0.5 mM TCEP, and 125 μg of copper PVT SPA beads (Perkin Elmer, Waltham, Massachusetts). Binding of 100 μM [$^{35}$S]O$_4$$^{2-}$ (at a specific activity of 50 mCi/mmol; mixed with non-labeled Na$_2$SO$_4$) by CysZ was measured in 96-well clear-bottom plates. For isotopic dilutions or competition assays, 10 nM – 100 mM of Na$_2$SO$_4$ or Na$_2$SO$_3$, respectively, were added simultaneously with the radiolabeled substrate. 800 mM imidazole were added to a set of samples (for each individual condition) to determine the non-proximity (background) signal as imidazole competes with His-tagged CysZ for binding to the Cu$^{2+}$-coated SPA beads. Plates were agitated for a minimum of 30 min or up to 16 hr at 4°C, and measured in the SPA mode of a Wallac MicroBeta1450 scintillation counter (Perkin Elmer, Waltham, Massachusetts). All data were analyzed with GraphPad Prism7 software.

## Microscale thermophoresis (MST)

Direct binding of SO$_3$$^{2-}$ by CysZ was measured with microscale thermophoresis using the Monolith NT.LabelFree (NanoTemper Technologies, Germany), which detects the relative change in native tryptophan emission at 336 nm. The binding affinity was determined using 500 nM purified CysZ from either *P. denitrificans* or *I. loihiensis* by mixing the protein with a serial dilution of Na$_2$SO$_3$ (0.5–47.5 mM) in buffer containing 20 mM Hepes, pH 6.5, 10% glycerol, 100 mM NaCl, 0.5 mM TCEP and 0.1% n-decyl-β-D-maltopyranoside (DM). The osmolality of all test solutions was maintained by adding appropriate concentrations of NaCl. The mixture was incubated for 20 min at room temperature and then loaded into Monolith NT.LabelFree Zero Background Standard Treated Capillaries. Measurements were carried out at high (60%) MST power and 15% excitation power using the MO. Control v1.4.4 software. The MST data at 19–20 s after IR laser power exposure was collected and

the normalized fluorescence was subjected to non-linear regression fitting in Prism seven to obtain the $EC_{50}$. MST measurements involving PfCysZ were inconclusive.

## Measurement of CysZ single-channel activity in the planar lipid bilayer

A previously described method (Mueller et al., 1962) to insert ion channels incorporated in liposomes into 'painted' planar lipid bilayers by vesicle fusion was used to incorporate CysZ into a lipid bilayer created on a small aperture between two aqueous compartments, called the cis and trans compartments (Morera et al., 2007). Phosphatidylethanolamine (PE) and phosphatidylserine (PS) (Avanti Polar Lipids, Alabaster, Alabama), in a ratio of 1:1 were dissolved in chloroform to mix, and dried completely under an argon stream. The mixed and dried lipids were then dissolved in n-decane to a final concentration of 50 µg/ml, and kept at 4°C. The lipids are always prepared fresh, on the same day of the experiment. The purified CysZ proteins were incorporated into PE:PS (1:1) liposomes by brief sonication at 80 kHz for 1 min at 4°C. Since this system is very sensitive to contaminants, CysZ was expressed in and purified from a porin-deficient strain of E. coli cells to prevent any carry through of contaminating porins that could create large conductances and artifacts in the single-channel recordings. However, we cannot rule out that our preparation contained such contaminants that eventually precluded more detailed electrophysiological measurements. The experimental apparatus consisted of two 1 ml buffer chambers separated by a Teflon film that contains a single 20- to 50 µm hole. A lipid bilayer was formed by 'painting' the hole with the 1:1 mixture of PE:PS, this results in a seal between the two cups formed by the lipids (Leal-Pinto et al., 1995). For these studies, the cis side was defined as the chamber connected to the voltage-holding electrode and all voltages are referenced to the trans (ground) chamber. Stability of the bilayer was determined by clamping voltage at various levels. If a resistance >100 Ω and noise <0.2 pA were maintained in the patch, the proteoliposomes containing CysZ were added to the trans side of the chamber and stirred for 1 min. The fusion event or insertion of a channel into the bilayer was assessed by the presence of clear transitions from 0 current to an open state.

## Site-specific cysteine labeling experiments

All functional and cysteine mutants of CysZ were generated by site-directed mutagenesis, using the QuikChange site-directed mutagenesis kit (Agilent Technologies, Santa Clara, California). The sequence-verified mutants were then tested for expression in comparison to the WT CysZ. To address the membrane topology of IlCysZ, single cysteine mutants were designed to perform site-directed fluorescence labeling based on the accessibility of the cysteine to the membrane impermeable thiol-directed fluorescent probe (Ye et al., 2001). A set of surface-exposed residues at different positions on the CysZ molecule were selected to be mutated to cysteines, based on the IlCysZ structure. The cysteine mutants were expressed by standard protocols that were used for the WT protein. The membrane fraction of each mutant was pelleted after cell lysis by ultracentrifugation at 100,000 x g and resuspended at 20 mg/ml (Bradford assay) in fresh buffer containing 20 mM Na-Hepes pH 7.0, 200 mM NaCl, protease inhibitors: 0.5 mM PMSF and Complete protease inhibitor cocktail EDTA-free and 1 mM TCEP-HCl. 1 ml of membranes were then incubated with 30 µM membrane-impermeant fluorescein-5-maleimide dye (2 mM stock freshly prepared in water, protected from light) for 30 min in the dark at room temperature. The labeling reaction was stopped by the addition of 6 mM β- mercaptoethanol, and the membranes were spun down and resuspended in fresh buffer to remove any remaining unreacted fluorescent dye. The fluorescently labeled protein was then purified from the membranes by solubilization with 1% DM, using a standard Ni-NTA purification protocol, qualitatively analyzed on an SDS-PAGE, and quantitatively measured by a Tecan fluorescence plate reader at an excitation of 495 nm, emission of 535 nm.

## Crosslinking of cysteine mutant pairs in CysZ

Cysteine mutants of PfCysZ and IlCysZ were designed based on pairs of residues that were in close proximity (within 3–7 Å) of each other in our structure that could have the ability to covalently join the 2 protomers of the dimer. The single and double cysteine mutants were made by site-directed mutagenesis using the QuikChange Site-Directed Mutagenesis Kit (Agilent Technologies, Santa Clara, California). Mutants and WT were expressed using the standard protocols CysZ expression. Isolated membrane fraction was resuspended by homogenization in fresh lysis buffer at a membrane

protein concentration of ~25 mg/ml (measured by Bradford Assay). Bis-methanethiosulfonate (Bis-MTS) crosslinkers (Santa Cruz Biotechnologies, Dallas, Texas) of different spacer lengths were used at 0.5 mM (dissolved in DMSO) added to 1 ml of resuspended membranes at RT for 1 hr. Bis-MTS crosslinkers are membrane permeable, highly reactive and specific to sulfhydryl groups, and the covalent linkage is resistant to reducing agents like β-mercaptoethanol (*Akabas et al., 1992*). The crosslinking lengths used were: 1,1-Methanediyl Bismethanethiosulfonate (3.6 Å), 1,2- Ethanediyl Bis-methanethiosulfonate (5.2 Å), 1,4-Butanediyl Bismethanethiosulfonate (7.8 Å) and 1,6-Hexanediyl Bismethanethiosulfonate (10.4 Å). The reaction was quenched by the addition of 10 mM free cysteine, and the protein was then extracted and purified from the membranes with 1% DM followed by Ni-NTA resin. The imidazole is then diluted out in the Ni-elute, and the His-tag is cut with TEV protease in small scale. The cleaved protein is passaged over Ni-NTAresin to remove contaminants and uncleaved protein, and flowthrough is run on a reducing SDS- PAGE and stained with Coomassie blue to analyze dimer formation. The mutants designed were L161C (single mutant), L161C-A164C and N160C-L168C (double mutants) for *Pf*CysZ; and L156C-Q163C and V157-Q163C (double mutants) for *Il*CysZ.

## PDB accession codes

*Idiomarina loihiensis* CysZ: 3TX3, *Pseudomonas fragi* CysZ: 6D79, *Pseudomonas denitrificans* CysZ: 6D9Z.

## Acknowledgements

Crystallographic data for this study were collected on the NSLS beamline X4A at Brookhaven National Laboratory (supported by the New York Structural Biology Center) and on the NE-CAT beamlines 24ID-C and E (supported by NIH-NIGMS grant P41 GM103403) at the Advanced Photon Source. The Pilatus 6M detector on 24-ID-C beam line is funded by a NIH-ORIP HEI grant (S10 RR029205). This work was supported by NIH-NIGMS grants R01 GM098617 (FM), R01 GM107462 (WAH) and R01 GM119396 (MQ), and by grants through the New York Structural Biology Center for the New York Consortium on Membrane Protein Structure (NYCOMPS; U54 GM095315) and for the Center on Membrane Protein Production and Analysis (COMPPÅ; P41 GM116799). OBC was supported by a Charles H Revson Senior Fellowship. We thank John Schwanof for assistance during data collection at NSLS, Jin Liang Sui for analysis of electrophysiology data at Northeastern University, Joe Mindell, Francisco Parra, Albano Meli, Giuliano Sciara and Carlos A Villalba Galea for helpful advice and contributions at the early stages of the project, and Leora Hamberger for her assistance managing the Mancia Lab.

## Additional information

### Funding

| Funder | Grant reference number | Author |
|---|---|---|
| National Institute of General Medical Sciences | R01 GM119396 | Matthias Quick |
| National Institute of General Medical Sciences | R01 GM107462 | Wayne A Hendrickson |
| National Institute of General Medical Sciences | U54 GM095315 | Wayne A Hendrickson |
| National Institute of General Medical Sciences | P41 GM116799 | Wayne A Hendrickson |
| National Institute of General Medical Sciences | R01 GM098617 | Filippo Mancia |
| New York Consortium on Membrane Protein Structure | NYCOMPS; U54 GM095315 | Filippo Mancia |

The funders had no role in study design, data collection and interpretation, or the decision to submit the work for publication.

## Author contributions

Zahra Assur Sanghai, Conceptualization, Data curation, Formal analysis, Validation, Investigation, Visualization, Methodology, Writing—original draft, Writing—review and editing; Qun Liu, Data curation, Formal analysis, Methodology, Writing—review and editing; Oliver B Clarke, Data curation, Formal analysis, Validation, Methodology, Writing—review and editing; Meagan Belcher-Dufrisne, Pattama Wiriyasermkul, Edgar Leal-Pinto, Data curation, Formal analysis, Methodology; M Hunter Giese, Conceptualization, Data curation, Formal analysis, Methodology; Brian Kloss, Shantelle Tabuso, James Love, Data curation, Methodology; Marco Punta, Surajit Banerjee, Kanagalaghatta R Rajashankar, Data curation, Formal analysis; Burkhard Rost, Conceptualization, Validation; Diomedes Logothetis, Formal analysis, Supervision, Validation, Investigation, Writing—review and editing; Matthias Quick, Conceptualization, Data curation, Formal analysis, Supervision, Validation, Methodology, Writing—review and editing; Wayne A Hendrickson, Conceptualization, Formal analysis, Supervision, Funding acquisition, Validation, Writing—original draft, Writing—review and editing; Filippo Mancia, Conceptualization, Formal analysis, Supervision, Funding acquisition, Validation, Writing—original draft, Project administration, Writing—review and editing

## Author ORCIDs

Oliver B Clarke https://orcid.org/0000-0003-1876-196X
Filippo Mancia https://orcid.org/0000-0003-3293-2200

## Decision letter and Author response

Decision letter https://doi.org/10.7554/eLife.27829.026
Author response https://doi.org/10.7554/eLife.27829.027

# Additional files

## Supplementary files

• Transparent reporting form
DOI: https://doi.org/10.7554/eLife.27829.018

## Data availability

Diffraction data has been deposited in the PDB under accession codes: Idiomarina loihiensis CysZ: 3TX3, Pseudomonas fragi CysZ: 6D79, Pseudomonas denitrificans CysZ: 6D9Z

The following datasets were generated:

| Author(s) | Year | Dataset title | Dataset URL | Database, license, and accessibility information |
|---|---|---|---|---|
| Zahra Assur Sanghai, Qun Liu, Wayne A Hendrickson, Filippo Mancia | 2011 | CysZ, a putative sulfate permease | https://www.rcsb.org/structure/3TX3 | Publicly available at the RCSB Protein Data Bank (accession no. 3TX3) |
| Zahra Assur Sanghai, Qun Liu, Oliver B Clarke, Surajit Banerjee, Kanagalaghatta R Rajashankar, Wayne A Hendrickson, Filippo Mancia | 2018 | Structure of CysZ, a sulfate permease from Pseudomonas Fragi | https://www.rcsb.org/structure/6D79 | Publicly available at the RCSB Protein Data Bank (accession no. 6D79) |
| Zahra Assur Sanghai, Oliver B Clarke, Qun Liu, Surajit Banerjee, Kanagalaghatta R Rajashankar, Wayne A Hendrickson, Filippo Mancia | 2018 | Structure of CysZ, a sulfate permease from Pseudomonas Denitrificans | https://www.rcsb.org/structure/6D9Z | Publicly available at the RCSB Protein Data Bank (accession no. 6D9Z) |

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
