## [Decision Letter]

Thank you for submitting your article "Structure-based analysis of CysZ-mediated cellular uptake of sulfate" for consideration by *eLife*. Your article has been reviewed by three peer reviewers, one of whom is a member of our Board of Reviewing Editors and the evaluation has been overseen by Richard Aldrich as the Senior Editor. The following individuals involved in review of your submission has agreed to reveal his identity: Ming Zhou (Reviewer #2).

The reviewers have discussed the reviews with one another and the Reviewing Editor has drafted this decision to help you prepare a revised submission.

Summary:

The reviewers have expressed shared enthusiasm for the series of crystal structures showing novel and unusual assembly of sulfate transporters/channels. However, several shared concerns have been raised in regard to the biochemical examination of the oligomerization assembly and functional studies. The key points are summarized below, and individual reviewers' comments are included.

Essential revisions:

1) The channel nature of the proteins is not convincingly demonstrated. The electrophysiology experiments are of insufficient quality. They should either be removed or improved. Specifically, the currents at higher protein concentrations, originating from multiple channels in bilayer should be shown and should have properties consistent with observed single channels. Additionally, the fact that 5µM concentrations of sulfite inhibit flux in the presence of 150 mM sulfate seems incompatible with comparable affinities determined in binding experiments. It should be demonstrated that sulfite inhibits sulfate currents in a dose-dependent and reversible manner.

2) The liposome uptake raises a question: How can uptake occur against the buildup of electrical potential? Furthermore, the apparent uptake rate seems too slow to be compatible with observed single channel activity. The authors should examine whether the process is electrogenic (as would be suggested by electrophysiology) by, for example, using K^+^/valinomycin system. Finally, reviewers were alarmed by very different sulfate uptake levels (including in liposome controls) shown in Figure 4D and Figure 4—figure supplement 1A.

3) Based on comments in (1) and (2), the reviewers are not convinced that the channel nature of these proteins has been convincingly demonstrated. This doubt is also increased because it seems unclear how sulfate would overcome the generally negative-inside potential of the bacterial cells that would be expected to oppose significant sulfate influx.

4) The uptake experiments in bacteria need to be controlled for protein expression.

5) The reviewers would also like to see whether all three proteins are able to bind sulfate and/or sulfite in ITC experiments.

6) The shown biochemical support for hexameric assembly is not sufficient. Cross-linking in all proteins across the second proposed interface would be required or perhaps cross-linking of the complete hexamer by, for example, glutaraldehyde.

7) In the absence of convincing functional data showing channel properties, proposal of a permeation mechanism might be premature.

Reviewer #2:

1) Overall, the discovery and validation that CysZ is a dual topology membrane protein are solid and novel. However, the proposed sulfate permeation pathway seems only compatible in PdCysZ. It is not clear whether there is a protein-enclosed central cavity in the middle of the membrane in IlCysZ and PfCysZ. Another way of looking at it is to ask whether the minimal functional unit of CysZs, is a dimer or hexamer. The authors went so far as to say that the two forms of dimers could be derived from the hexamers but stayed short of saying whether IlCysZ and PfCysZ were hexamers to begin with. Are there any data on whether a dimer is sufficient for the transport function?

2) On the functional side, the cell-based assays confirmed and advanced the previous notion of CysZ functions, and the binding and transport assays on the purified proteins added more precision to our understanding of the protein. For the transport assay shown in Figure 4—figure supplement 1, there are about 20 turnovers in 60 seconds if one assumes a hexameric basic functional unit, and about 8 turnovers if a dimer is a basic functional unit (Figure 4—figure supplement 1A). Although there is nothing wrong with the relatively slow rate of transport, the slow rate is not compatible with the high turnover rate required for observing single channel activities in patch clamp experiments shown in Figure 4—figure supplement 2.

3) Finally, reading through the manuscript, it seems that a transmembrane pore overlapping with the three-fold axis of the hexamer would not be out of the question? The authors mentioned this possibility but seem to dismiss it right away mainly because the tip of the H4b-H5a is not highly conserved. However, since the structures already showed that this unit is mobile, the tip may not end up coordinating the incoming sulfate after all.

Reviewer #3:

1) Biochemical experiments do not independently support dual topology, contrary to the authors' argument. Independent biochemical support for dual topology assembly is overstated (subsection “Dual-topology assembly of CysZ”). The crosslinking experiments support the dimerization arrangement seen in the structure, but I am not convinced that they rule out other assemblies. The cysteines are located at a position thought to be right in the middle of the membrane. Thus, the cysteines in a parallel subunit would be positioned at a similar depth and might also crosslink.

The authors argue that the cysteine accessibility experiments shown in Figure 3—figure supplement 2 support a dual-topology model (subsection “Dual-topology assembly of CysZ”). They do not. "Sidedness" is lost when the membranes are isolated.

A straightforward analysis of positive charge distribution on the loops might provide complementary support for physiological dual topology assembly, since the structure provides good support for the location of the loops, and bacteria tend to follow well-known rules for membrane protein insertion.

2) The authors do not state what density of reconstituted protein is used for electrophysiology; if they use the same density of protein as they do in their radioactive uptake experiments, then they ought to see multiple channels incorporating into the bilayer with each liposome fusion if most of the protein is functional. Do they ever see multichannel currents? How many independent times have they seen single-channel insertions with similar characteristics?

3) The authors do not show the Se anomalous difference map for SeO4-, or the Fo-Fc omit maps for the SO4-bound and mutant structures to support their identification of the substrate binding site (subsection “Sulfate Binding Site”).

4) I would suggest using "antiparallel" rather than "dual topology" to describe the assembly. I think "antiparallel" is a slightly more precise term, since "dual topology" refers to stable subunit insertion in membranes. It's hard to imagine these monomers existing happily without a partner in the membrane.

[Editors' note: further revisions were requested prior to acceptance, as described below.]

Thank you for resubmitting your work entitled "Structure-based analysis of CysZ-mediated cellular uptake of sulfate" for further consideration at *eLife*. Your revised article has been favorably evaluated by Richard Aldrich (Senior editor), and a Reviewing editor.

The manuscript has been improved but there are some remaining issues that need to be addressed before acceptance, as outlined below:

The functional experiments have been substantially improved. Still, some doubts remain that we would like to have clarified. We are still not convinced that CysZ is a channel. Primarily, we still do not understand how any substantial accumulation of sulfate in liposomes could occur in the absence pf K(out)/valinomycin or Na(out)/Gramicidin system to counteract rapid buildup of negative-inside potential from sulfate uptake. I would need to see the following estimates from the authors:

1) In experiment shown in Figure 4B (SO4(2-) uptake over time), what concentration of SO4(2-) in the liposomes is achieved at the maximum of uptake? Is it below/above the external SO4(2-)? What electric potential is expected to be established at this point if CysZ is sulfate-specific channel.

2) To compare liposome-based assays to single channel recordings, the authors make parallels to Chris Miler's paper on Rb flux into K channel-containing liposomes. Those experiments were set up differently from the current experiments: K/Rb exchange was measured at microM concentrations of external Rb. Linearity of Rb uptake rates from microM to high mM was assumed (if we understand correctly). In the current experiments, it is not exchange that is measured, and it is not clear what determined the plateau in Figure 4B. Moreover. the Km for sulfate is expected to be ca 3 mM and linearity up to 150 mM could not be assumed. We would like to see more detailed reasoning from the authors on how the liposome-based rates relate to single channel conductance.

[Editor’s note: further revisions were requestged prior to acceptance, as described below.]

The editors have some questions regarding your latest resubmission that might be easier to address in a dialogue format. They are still concerned with the statement that CysZ is an SO4(2-) channel, especially if it is only permeant to anions. As you mentioned in your response, there might be sodium ions co-permeating with sulfate or something else is going on. In fact, there has to be! The process of sulfate accumulation in the liposome must be electroneutral (or nearly so) or you would not see it. Imagine that it is not, then bringing every sulfate molecule in would mean bringing two extra negative charges. You can calculate how much opposing negative inside potential that would generate by using the relationship E = Q/C, where Q is the extra charge that you bring in and C is the liposome membrane capacitance (See Hille's book, Chapter 1). Do you agree? If so, bringing in 2.5 mM sulfate final internal concentration would result in something like -0.6 V potential. That is a huge potential! Thus, the process that you observe must be electroneutral. This is also confirmed by the very weak dependence on trans-membrane potential established using K gradient/valinomycin system. Incidentally, I don't think that using Nernst equation to calculate potential buildup as you do in your last response is correct.

---

## [Author Response]

Summary:The reviewers have expressed shared enthusiasm for the series of crystal structures showing novel and unusual assembly of sulfate transporters/channels. However, several shared concerns have been raised in regard to the biochemical examination of the oligomerization assembly and functional studies. The key points are summarized below, and individual reviewers' comments are included.We appreciate the enthusiasm shared by the reviewers, and the criticisms made to our work, which we interpreted in the most positive of ways. We respond to each point in detail, below.Essential revisions:1) The channel nature of the proteins is not convincingly demonstrated. The electrophysiology experiments are of insufficient quality. They should either be removed or improved. Specifically, the currents at higher protein concentrations, originating from multiple channels in bilayer should be shown and should have properties consistent with observed single channels. Additionally, the fact that 5µM concentrations of sulfite inhibit flux in the presence of 150 mM sulfate seems incompatible with comparable affinities determined in binding experiments. It should be demonstrated that sulfite inhibits sulfate currents in a dose-dependent and reversible manner.

We thank the reviewers for pointing out this inconsistency. First, we recognize that the reported electrophysiology experiments were not of high quality. Since we were unable to perform new electrophysiology experiments, we removed most of this description; however, we have retained representative current traces (Figure 4—figure supplement 1D), including a trace showing multiple levels of single-channel currents as suggested. We have also collected an entirely new set of proteoliposome-based radiolabeled sulfate flux measurements (see below), including extensive tests with ionophores that can disrupt transmembrane gradients, whereby we exclude likely transport partners such as protons or sodium ions. Thus, together with testing the effect of the membrane potential on sulfate flux (see below), the new proteoliposome results provide further support for our originally proposed channel-like behavior of CysZ. Superficially, however, currents from CysZ electrophysiology recordings seem at odds with much lower rates of sulfate flux measured in proteoliposomes. On the other hand, this apparent inconsistency was also reported for the well-studied KcsA where a comparison between concentrative radiotracer (^86^Rb^+^) fluxes into liposomes and electrophysiological bilayer measurements revealed similar rate constants (Heginbotham, Kolmakova-Partenskh and Miller, 1998). We have performed the same type of analysis and found that the situation here is qualitatively like that for KcsA; however, we decided to refrain from presenting these data in the manuscript as the calculations rely in major part on estimates and thus preclude an accurate quantitative assessment.

2) The liposome uptake raises a question: How can uptake occur against the buildup of electrical potential? Furthermore, the apparent uptake rate seems too slow to be compatible with observed single channel activity. The authors should examine whether the process is electrogenic (as would be suggested by electrophysiology) by, for example, using K^+^/valinomycin system. Finally, reviewers were alarmed by very different sulfate uptake levels (including in liposome controls) shown in Figure 4D and Figure 4—figure supplement 1A.3) Based on comments in (1) and (2), the reviewers are not convinced that the channel nature of these proteins has been convincingly demonstrated. This doubt is also increased because it seems unclear how sulfate would overcome the generally negative-inside potential of the bacterial cells that would be expected to oppose significant sulfate influx.4) The uptake experiments in bacteria need to be controlled for protein expression.

Critique points 2 – 4 directly connect to the concerns raised in point 1, and we have tackled these inconsistencies by performing an exhaustive set of experiments that addresses all points raised by the reviewers. In this context, we have repeated (and extended) all radiotracer-based uptake studies. Here, we have performed the kinetic characterization of CysZ-mediated sulfate flux in the proteoliposome system using known amounts of protein (1) to rule out that observed activity patterns reflect differences in the expression levels of CysZ rather than functional properties of CysZ as mentioned in critique point 4, and (2) to provide a suitable experimental platform for the measurements suggested by the reviewers for(a) the detailed kinetic characterization of sulfate flux, (b) measuring the inhibition of sulfite on sulfate flux, (c) elucidating the effect of electrochemical ion gradients on CysZ activity, and d) analyzing the effect of the membrane potential on CysZ-mediated sulfate flux to provide ‘electrophysiology-like’ measurements of this system in the membrane. To connect our studies to those performed by Para et al., 1983 that first described the CysZ system, we still show sulfate uptake in *E. coli* wild-type cells compared to uptake measured in CysZ- *E. coli* cells (new Figure 4A). The purpose of showing this initial data set is to highlight that CysZ is not the only sulfate translocating system in these bacteria but that the other sulfate-translocating systems play a role in the sulfate homeostasis in bacteria.

All new flux studies were performed in CysZ-containing proteoliposomes, using similar experimental conditions as those applied to the cell-based studies. Employing proteoliposomes resulted in much ‘cleaner’ data compared to the initially shown cell-based uptake studies (new Figure 4B) and allowed us to determine the catalytic turnover number for all three CysZ isoforms. Furthermore, the proteoliposome system simplified the characterization of potential energizing co-factors in our experimental system as – in contrast to cell-based studies – the effect of native systems found in the bacterial membrane could be ruled out for our measurements. To test the effect of a potential proton or sodium driving force on CysZ activity, we used gramicidin to dissipate the proton and sodium electrochemical gradient across the membrane, thus providing a more direct assessment of the ion requirements rather than working with uncouplers that target proton-translocating ATPases in the bacterial membrane as shown in the original submission of this manuscript. Based on these findings, we could address the concerns raised in Critique Points 2 and 3 and follow the suggested experiment involving a potassium flux-generated membrane potential of defined polarity by the addition of the potassium ionophore valinomycin. We observed an increase in sulfate flux when we applied depolarizing conditions (inside positive) in the CysZ-containing proteoliposomes, a finding that directly addresses the issue raised by the reviewers regarding the buildup of a negative potential as a consequence of the accumulation of the negatively charged sulfate ion. However, our data also revealed that CysZ can mediate the flux of sulfate in inside-negative proteoliposomes or those that exhibit no membrane potential. We infer from studies performed in context with the characterization of potassium channels in membraneous systems (including proteoliposomes) by Chris Miller’s lab that the gradient of the transported ion (here sulfate) serves as driving force that acts against the buildup of electrical charge over the course of the flux experiment. Under physiological conditions, as outlined in Figure 1—figure supplement 1, CysZ is part of an operon that is involved in the synthesis of Cys and the incoming sulfate is quickly assimilated in this process, thus preventing a buildup of negative charge in the cells. In this context, it is worth noting that the sulfate flux mediated by CysZ is rather quick and peaks at about 10 – 15 seconds, followed by a decrease to about 50% within 90 seconds (new Figure 4B). This observation shows that the charge buildup in the proteoliposomes actually acts against the transmembrane sulfate gradient as the uptake curve lacks a steady state of sulfate accumulation as expected for a process that is energized by an additional component (i.e., the concentration gradient of a second ion found in secondary active transport processes).

5) The reviewers would also like to see whether all three proteins are able to bind sulfate and/or sulfite in ITC experiments.

This is a critical point in our understanding of whether sulfite actually binds to CysZ and we thank the reviewers for pointing out this shortcoming in our initial submission. We have addressed this issue by performing a new set of experiments utilizing microscale thermophoresis (MST). MST detects changes in the hydration shell of biomolecules along an applied temperature gradient in response to conformational changes of the biomolecules, allowing for a quantification of ligand binding events, without requiring surface immobilization and high sample consumption (see e.g., Seidel et al., 2012). We have performed, and now show in Figure 4—figure supplement 1D, SO_3_^2-^ binding by *Pd*CysZ and *Il*CysZ. The *EC_50_* values for SO_3_^2-^ binding based on the MST measurements is essentially in perfect agreement with the *IC_50_* values determined by blocking radiolabeled sulfate binding with increasing concentrations of non-labeled sulfite. Only our attempts to measure SO_3_^2-^ binding by *Pf*CysZ produced inconclusive data due to high background signals. However, all our SPA-based radiolabeled sulfate binding measurements were also repeated for the revision of the manuscript. In the course of performing these experiments, we realized that the initially reported SPA-based binding studies did not necessarily reflect the CysZ-specific activity but appeared to be a mix of CysZ-specific properties and elevated background (non-proximity) signals. In a very time- and effort-consuming process we were able to determine that several batches of the copper-coated YSi SPA beads bound sulfate even in the absence of protein at a level that interfered with the CysZ-specific measurements. After contacting the manufacturer of the SPA beads (Perkin Elmer) and sorting out defective batches/vials of the SPA beads, we are happy to be able to report a comprehensive set of binding studies in this revision.

6) The shown biochemical support for hexameric assembly is not sufficient. Cross-linking in all proteins across the second proposed interface would be required or perhaps cross-linking of the complete hexamer by, for example, glutaraldehyde.

We performed numerous experiments to address this question. These included (i) cross-linking with glutaraldehyde and another amino-reactive cross-linker on *E. coli* membranes expressing various CysZ orthologs, (ii) cross-linking with glutaraldehyde on CysZ proteins reconstituted in liposomes after purification, (iii) negative stain electron microscopy on protein in detergent and reconstituted in lipid-filed nanodiscs, and (iv) cryo-electron microscopy on protein in nanodiscs and in proteoliposomes. Unfortunately, and frustratingly, the results for all of these experiments were inconclusive. It has now become obvious that optimizing these approaches would require a considerable amount of additional time, beyond what we consider acceptable. However, all the plethora of functional experiments (previous and new) presented here, show that the three CysZs for which we have structures, have essentially indistinguishable functional properties and profiles. The functional equivalence of all CysZ, together with the fact that the it is implausible that the CysZ monomer be the functional unit, that the two CysZ dimeric assemblies (*Il*CysZ and *Pf*CysZ) observed in our structures are unrelated, and that they are instead both present in the CysZ hexameric assembly (*Pd*CysZ), strongly supports our claim that the hexamer is the functional unit of CysZ.

7) In the absence of convincing functional data showing channel properties, proposal of a permeation mechanism might be premature.

We agree with this criticism that was based on our initial functional studies. We anticipate that the new set of functional studies unequivocally support our conclusion about the channel-like properties of CysZ.

Reviewer #2:1) Overall, the discovery and validation that CysZ is a dual topology membrane protein are solid and novel. However, the proposed sulfate permeation pathway seems only compatible in PdCysZ. It is not clear whether there is a protein-enclosed central cavity in the middle of the membrane in IlCysZ and PfCysZ. Another way of looking at it is to ask whether the minimal functional unit of CysZs, is a dimer or hexamer. The authors went so far as to say that the two forms of dimers could be derived from the hexamers but stayed short of saying whether IlCysZ and PfCysZ were hexamers to begin with. Are there any data on whether a dimer is sufficient for the transport function?

See point 6 above.

2) On the functional side, the cell-based assays confirmed and advanced the previous notion of CysZ functions, and the binding and transport assays on the purified proteins added more precision to our understanding of the protein. For the transport assay shown in Figure 4—figure supplement 1, there are about 20 turnovers in 60 seconds if one assumes a hexameric basic functional unit, and about 8 turnovers if a dimer is a basic functional unit (Figure 4—figure supplement 1A). Although there is nothing wrong with the relatively slow rate of transport, the slow rate is not compatible with the high turnover rate required for observing single channel activities in patch clamp experiments shown in Figure 4—figure supplement 2.

We appreciate the reviewer’s comment about the relatively slow turnover number of CysZ compared to other channel proteins. We have mentioned in the main text that we cannot rule out that the activity of CysZ, in fact, may reflect the energetically uncoupled flux of SO_4_^2-^ along its concentration gradient, characteristic for a low-turnover channel phenotype (or passive transport/uniport, or facilitated diffusion) and cite a paper in this context that addresses the blurry boundaries between channels and transporters (Ashcroft et al., 2008). From a mechanistic standpoint, however, our data clearly support an uncoupled translocation mechanism of sulfate (reflected in the term ‘channel-like’ behavior).

3) Finally, reading through the manuscript, it seems that a transmembrane pore overlapping with the three-fold axis of the hexamer would not be out of the question? The authors mentioned this possibility but seem to dismiss it right away mainly because the tip of the H4b-H5a is not highly conserved. However, since the structures already showed that this unit is mobile, the tip may not end up coordinating the incoming sulfate after all.Reviewer #3:1) Biochemical experiments do not independently support dual topology, contrary to the authors' argument. Independent biochemical support for dual topology assembly is overstated (subsection “Dual-topology assembly of CysZ”). The crosslinking experiments support the dimerization arrangement seen in the structure, but I am not convinced that they rule out other assemblies. The cysteines are located at a position thought to be right in the middle of the membrane. Thus, the cysteines in a parallel subunit would be positioned at a similar depth and might also crosslink.The authors argue that the cysteine accessibility experiments shown in Figure 3—figure supplement 2 support a dual-topology model (subsection “Dual-topology assembly of CysZ”). They do not. "Sidedness" is lost when the membranes are isolated.A straightforward analysis of positive charge distribution on the loops might provide complementary support for physiological dual topology assembly, since the structure provides good support for the location of the loops, and bacteria tend to follow well-known rules for membrane protein insertion.A positive charge analysis shows that there are many more positively charged residues on the side of the hydrophilic head of CysZ than on the other side where only one loop between helices 2b and 3a, containing no positive charges, is present, implying that CysZ should orient in the membrane with its hydrophilic head on the cytoplasmic side. Although we agree with the reviewer, that sidedness is indeed lost when membranes are isolated, we believe that our disulfide crosslinking experiments confirm that a dimer crosslinked between residues L161 and A164 (Figure 3—figure supplement 2) might only form if the 2 protomers are in an antiparallel conformation within the membrane.2) The authors do not state what density of reconstituted protein is used for electrophysiology; if they use the same density of protein as they do in their radioactive uptake experiments, then they ought to see multiple channels incorporating into the bilayer with each liposome fusion if most of the protein is functional. Do they ever see multichannel currents? How many independent times have they seen single-channel insertions with similar characteristics?

As mentioned above, we have decided to remove the majority of the electrophysiological measurements that, as the reviewer pointed out here, were not controlled for protein density and do, in fact, seem to reflect multichannel currents.

3) The authors do not show the Se anomalous difference map for SeO4-, or the Fo-Fc omit maps for the SO4-bound and mutant structures to support their identification of the substrate binding site (subsection “Sulfate Binding Site”).

We have now included an additional supplementary figure (Figure 5—figure supplement 1), to show the anomalous difference map highlighting the sulfate binding site residues, with density shown for the bound sulfate ion, selenate ion as well as water molecule (in the Apo structure).

4) I would suggest using "antiparallel" rather than "dual topology" to describe the assembly. I think "antiparallel" is a slightly more precise term, since "dual topology" refers to stable subunit insertion in membranes. It's hard to imagine these monomers existing happily without a partner in the membrane.

We thank the reviewer for his/her comments and agree that ‘antiparallel’ is indeed more appropriate to describe the CysZ assembly and have made the changes in the text.

[Editors' note: further revisions were requested prior to acceptance, as described below.]

The functional experiments have been substantially improved. Still, some doubts remain that we would like to have clarified. We are still not convinced that CysZ is a channel. Primarily, we still do not understand how any substantial accumulation of sulfate in liposomes could occur in the absence pf K(out)/valinomycin or Na(out)/Gramicidin system to counteract rapid buildup of negative-inside potential from sulfate uptake. I would need to see the following estimates from the authors:

We appreciate the concern, and we hope that our responses to the specific concerns below and the associated modifications in the manuscript will have addressed the issues appropriately.

1) In experiment shown in Figure 4B (SO4(2-) uptake over time), what concentration of SO4(2-) in the liposomes is achieved at the maximum of uptake? Is it below/above the external SO4(2-)? What electric potential is expected to be established at this point if CysZ is sulfate-specific channel.

We have performed the calculations for the SO_4_^2-^ accumulation in CysZ-containing liposomes depicted in Figure 4B. In an attached appendix, we have listed the individual steps of the calculation for the CysZ-mediated SO_4_^2-^ flux. We then compare these calculations for sulfate uptake by CysZ with similar calculations with those for tryptophan uptake by a secondary active transporter (MhsT). The MhsT experiments were performed in virtually identical reconstitution conditions as those used for CysZ, which allows for a direct side-by-side comparison.

Our calculations reveal that the accumulation of SO_4_^2-^ in liposomes containing CysZ slightly exceeds equilibrium, but that this level is dramatically lower than the accumulation observed for a known transporter, thus supporting a channel-like phenotype (as facilitator or uniporter). Considering inaccuracies in the calculations, the same bias would apply for the CysZ calculations and the MhsT calculations. We note that it is difficult to explain our observations if CysZ is assumed to be perfectly selective for SO_4_^2-^ as that condition would require that the internal concentration of SO_4_^2-^ always be less than the external concentration. In light of this, it seems feasible to consider that CysZ is permeable to other ionic species with Na^+^ being the most likely co-permeant ion for the conditions used in our experiments. Accordingly, we have indicated this in the text. We now include the conclusion about the uptake level for CysZ in comparison with the MhsT transporter (Malinauskaite et al., 2014) in the main text, and we have added the Calculations Appendix in the Supplemental Information section of the manuscript.

2) To compare liposome-based assays to single channel recordings, the authors make parallels to Chris Miler's paper on Rb flux into K channel-containing liposomes. Those experiments were set up differently from the current experiments: K/Rb exchange was measured at microM concentrations of external Rb. Linearity of Rb uptake rates from microM to high mM was assumed (if we understand correctly). In the current experiments, it is not exchange that is measured, and it is not clear what determined the plateau in Figure 4B. Moreover. the Km for sulfate is expected to be ca 3 mM and linearity up to 150 mM could not be assumed. We would like to see more detailed reasoning from the authors on how the liposome-based rates relate to single channel conductance.

The editor is indeed correct. The Rb^+^ flux measurements of Heginbotham et al., (1998) reflect the exchange of pre-loaded internal K^+^ and radiotracer amounts of external Rb^+^, and linearity of the rates from microM to milliM could be correctly assumed. Given the fact that the K_M_ of SO_4_^2-^ flux by CysZ is in the 3 mM range, the statement in the re-submitted manuscript about the linearity of rates up to 150 mM (conditions in the electrophysiological bilayer measurements) was – as the editor pointed out – not correct. It was our intention merely to provide a rationale for reconciling the apparent rates of sulfate uptake as measured by electrophysiology and by flux measurements, not to provide a direct comparison between rates determined by the two methods. Nevertheless, we realize that our statement assuming linearity is misleading. In fact, after re-assessing our SO_4_^2-^ flux experiments, we realized that increasing concentrations of sulfate used for the determination of the catalytic turnover number (Figure 4C) would increase the internal electrical potential build-up in a concentration-dependent manner. Thus, it is feasible to assume that at high sulfate concentrations the charge build-up would inhibit the accumulation of sulfate, leading to a significant underestimation of the actual flux rates at higher SO_4_^2-^ concentrations. Since the 5-s uptake measurements were already experimentally challenging, shorter time points (to capture the real initial rate of accumulation) could not be sampled. As a consequence, the actual accumulation of sulfate at high concentrations (that show a plateau in the concentration-dependent uptake curve) may be underestimated and thus lead to an underestimated catalytic turnover number. Since this ‘problem’ can be circumvented in the electrophysiological bilayer experiments, we realize that the comparison of rates assessed with these two different methods is not necessarily useful and we have clarified this in the revised manuscript.

[Editor’s note: further revisions were requested prior to acceptance, as described below.]

The editors have some questions regarding your latest resubmission that might be easier to address in a dialogue format. They are still concerned with the statement that CysZ is an SO4(2-) channel, especially if it is only permeant to anions. As you mentioned in your response, there might be sodium ions co-permeating with sulfate or something else is going on. In fact, there has to be! The process of sulfate accumulation in the liposome must be electroneutral (or nearly so) or you would not see it. Imagine that it is not, then bringing every sulfate molecule in would mean bringing two extra negative charges. You can calculate how much opposing negative inside potential that would generate by using the relationship E = Q/C, where Q is the extra charge that you bring in and C is the liposome membrane capacitance (See Hille's book, Chapter 1). Do you agree? If so, bringing in 2.5 mM sulfate final internal concentration would result in something like -0.6 V potential. That is a huge potential! Thus, the process that you observe must be electroneutral. This is also confirmed by the very weak dependence on trans-membrane potential established using K gradient/valinomycin system. Incidentally, I don't think that using Nernst equation to calculate potential buildup as you do in your last response is correct.

Thank you for your email and suggestions, following which we have prepared a revised version of the manuscript which we hope satisfies the criteria for publication in eLife. Furthermore, the Appendix will be removed from our submission.